# Meta Context Engineering via Agentic Skill Evolution

**Haoran Ye** [1]  **Xuning He** [1]  **Vincent Arak** [2]  **Haonan Dong** [1]  **Guojie Song** [1]

## Abstract

The operational efficacy of large language models relies heavily on their inference-time context. This has established Context Engineering (CE) as a formal discipline for optimizing these inputs. Current CE methods rely on manually crafted harnesses, such as rigid generation-reflection workflows and predefined context schemas. They impose structural biases and restrict context optimization to a narrow, intuition-bound design space. To address this, we introduce **Meta Context Engineering (MCE)**, a bi-level framework that supersedes static CE heuristics by co-evolving *CE skills* and *context artifacts*. In MCE iterations, a meta-level agent refines engineering skills via *agentic crossover*, a deliberative search over the history of skills, their executions, and evaluations. A base-level agent executes these skills, learns from training rollouts, and optimizes context as flexible files and code. We evaluate MCE across five disparate domains under offline and online settings. MCE demonstrates consistent performance gains, achieving 5.6–53.8% relative improvement over state-of-the-art agentic CE methods (mean of 16.9%), while maintaining superior context adaptability, transferability, and efficiency in both context usage and training. Code is available at https://github.com/henry-yeh/mce.

## 1. Introduction

The operational efficacy of large language models (LLMs) is governed by the curation and orchestration of their inference-time context (Mei et al., 2025). As LLM infrastructure evolves from monolithic chat interfaces toward compound agentic systems, context optimization has become both a critical bottleneck and a powerful lever for its enhancement. This establishes the discipline of Context Engineering (CE), which focuses on the principled LLM context optimization to maximize downstream utility and enable continuous self-improvement (Hua et al., 2025).

Recent advances demonstrate CE's efficacy across diverse objectives, such as enhancing vertical-domain performance (Zhang et al., 2026; Cai et al., 2025b; Agrawal et al., 2025), solving long-horizon tasks (Zhang et al., 2025a; Sun et al., 2025; Hu et al., 2025a; Kang et al., 2025), coordinating multiple agents (Yuksekgonul et al., 2025; Wu et al., 2024a; Ma et al., 2026), and self-evolution driven by experience (Ye et al., 2024; Lu et al., 2026; Wei et al., 2025; Gao et al., 2025). Unlike optimizing LLM parameters, optimizing their context provides distinct methodological advantages: (1) *interpretability*, by encoding experience in natural language rather than opaque weights; (2) *efficiency*, by enabling rapid deployment without costly updates of model parameters; (3) *modularity*, facilitating the composition and transfer of established contexts; and (4) *robustness*, ensuring immunity to catastrophic forgetting by decoupling capability acquisition from model weights.

However, current CE systems are fundamentally constrained by manually crafted agentic harnesses that impose characteristic inductive biases. *At the context representation level*, different context structures impose their own trade-offs: case-based trajectories retain rich episodic traces but lack generalization (Zhou et al., 2025); itemized lists accumulate abstract insights but remain flat and structurally inexpressive (Zhang et al., 2026); and graph-based hierarchies offer flexible organization but incur high latency without consistently outperforming naive retrieval (Xu et al., 2025a; Ai et al., 2025). *At the context optimization level*, existing methods exhibit opposing yet equally limiting biases. On one hand, prompt-rewriting approaches such as GEPA (Agrawal et al., 2025) favor brevity, iteratively refining concise, high-level rules that fail in tasks requiring detailed strategies and deep domain knowledge (Zhang et al., 2026). On the other hand, additive-curation approaches (Zhang et al., 2026; Cai et al., 2025b; Suzgun et al., 2025; Cai et al., 2025a) favor verbosity, accumulating context through generation-reflection-curation pipelines that produce noisy context update (due to limited global visibility), incur excessive overhead (due to instance-

---

[1]State Key Laboratory of General Artificial Intelligence, School of Intelligence Science and Technology, Peking University, Beijing, China [2]School of Electronics Engineering and Computer Science, Peking University, Beijing, China. Correspondence to: Guojie Song <gjsong@pku.edu.cn>.

*Proceedings of the 43rd International Conference on Machine Learning*, Seoul, South Korea. PMLR 306, 2026. Copyright 2026 by the author(s).

level optimization), and cause context bloat (due to additive accumulation without holistic synthesis). Ultimately, these heuristic choices restrict CE to a narrow design subspace, precluding the discovery of task-optimal strategies that lie beyond human intuition.

To address these limitations, we introduce **Meta Context Engineering (MCE)**, a framework that supersedes static heuristics through the co-evolution of *CE skills* and *context artifacts*. MCE formalizes CE as a bi-level optimization problem, effectively decoupling the engineering strategy (how to represent and optimize context) from the resulting engineered artifact (what context is learned). *At the meta-level*, we propose *agentic skill evolution*, in which an agent iteratively refines CE skills (Anthropic, 2025): executable instructions and code that govern the CE process. This evolution is driven by agentic crossover, an evolutionary operator that synthesizes superior skills by reasoning across task specifications, historical CE trajectories, and performance metrics. *At the base-level*, an agent executes these evolved skills to learn from training rollouts and constructs context, in a fully agentic manner. Unlike prior methods that predefine context schemas, our base-agent leverages coding toolkits and file system access to instantiate and optimize context as flexible, programmatic artifacts.

In this manner, MCE replaces heuristic scaffolding with a generic design space. Prior state-of-the-art (SOTA) methods, such as the generation-reflection-curation workflow of agentic context engineering (ACE) (Zhang et al., 2026), represent singular points within the vast manifold of possible CE skills. By granting AI the full agency and capabilities to generate arbitrary code, invoke other LLMs, and manipulate file structures, MCE can reconstruct such pipelines, discover novel CE architectures, and dynamically adjust CE strategies.

We evaluate MCE across five diverse domains (finance, chemistry, medicine, law, and AI safety), using four LLMs, and against SOTA CE methods. Under offline and online settings respectively, MCE achieves 89.1% and 74.1% average relative improvement over the DeepSeek-V3.1 base model, outperforming the prior SOTA by 18.4% and 33.0%. In addition, MCE demonstrates superior 1) *context adaptability*: flexibly adjusting context length from 1.5K to 86K tokens across tasks, free from the brevity or verbosity biases of prior methods; 2) *context efficiency*: achieving better performance with fewer context tokens; 3) *context transferability*: exhibiting 4–7% lower performance degradation when transferring contexts from strong to weak models; and 4) *training efficiency*: accelerating training by $13.6\times$ and requiring $4.8\times$ fewer rollouts than ACE to achieve higher training accuracy.

**Conflict of Interest Disclosure.** The authors declare no financial conflicts of interest related to this work.

## 2. Background and Related Work

### 2.1. Agentic Context Engineering

While LLMs possess strong zero-shot capabilities, post-deployment context engineering (CE) is crucial for domain adaptation and self-improvement (Khattab et al., 2023; Hu et al., 2025b; Fang et al., 2025; Shinn et al., 2023; Yuksekgonul et al., 2025). Current SOTA CE methods rely on manually designed agentic harnesses: models accumulate experience through exploration-reflection workflows to update contexts with predefined schemas.

However, these harnesses impose characteristic inductive biases. *At the context representation level*, the trade-offs are distinct: case-based trajectories retain rich episodic traces but lack generalization and abstraction (Zhou et al., 2025; Wang et al., 2024b); itemized lists accumulate abstract insights but remain flat and structurally limited (Zhang et al., 2026; Suzgun et al., 2025); and hierarchical, graph-based memories incur high latency and complexity without consistently outperforming naive retrieval (Chhikara et al., 2025; Ai et al., 2025; Li et al., 2025; Xu et al., 2025a). *At the context optimization level*, existing methods exhibit opposing yet equally limiting biases. On one hand, prompt-rewriting approaches such as GEPA (Agrawal et al., 2025) employ full LLM-based rewrites driven by reflections over sampled trajectories, favoring abstract, high-level rules over detailed domain knowledge (brevity bias). On the other hand, additive-curation approaches such as Dynamic Cheatsheet (DC) (Suzgun et al., 2025) and Agentic Context Engineering (ACE) (Zhang et al., 2026) orchestrate modular agents (generators, reflectors, and curators) to iteratively perform on-policy rollouts, textual reflections, and context updates. These methods utilize localized updates to itemized lists and can additively accumulate insights, but suffer from structural rigidity and context bloat (Cai et al., 2025b;a).

We posit that no single agentic harness is universally optimal, motivating MCE's dual-level optimization. *Our instantiation of MCE is driven by two converging trends.* First, agent architectures are shifting from rigid, multi-agent scaffolds toward unified, self-looping, and self-spawning frameworks with maximal agency and minimal yet general tools (Anthropic, 2025a; Manus, 2025; langchain-ai, 2025; Bolin, 2026). In this architecture, domain specificity can be encapsulated in agent skills: organized instructions, scripts, and resources that agents discover and load dynamically (Anthropic, 2025). *This motivates our meta-level design: CE as a fully agentic process without restrictive scaffolding, with task specificity injected through learned skills.*

Second, coding toolkits and computer (file system) access

have became the essential harness of both general and vertical agents (Yang et al., 2025; Anthropic, 2026; Manus, 2025; Cheng et al., 2026). The prevalence stems largely from the versatility of Turing-complete programming languages, which offer maximal design flexibility (Zhang et al., 2025c;d), and their inherent verifiability, which facilitates robust training (Wei, 2025; Wang et al., 2024a; Yang et al., 2024). *This insight informs our base-level design space: context artifacts consist of files and code, which are general, agent-native representations unconstrained by predefined schemas.*

Ultimately, MCE represents a transition from manually crafted CE workflows to fully agentic meta- and self-learning systems, where domain specificity encapsulated with learned skills and context is managed via agent-generated files and code.

## 2.2. Evolutionary Computation with LLMs

Conventional evolutionary computation (EC) relies on manually designed operators, which struggle to capture complex solution structures and domain properties. Recent research demonstrates that LLMs can serve as intelligent genetic operators, leveraging semantic knowledge to generate meaningful variations without explicit rules (Wu et al., 2024b).

This paradigm shift enables the exploration of unstructured search spaces and facilitates optimization across varying levels of abstraction (Novikov et al., 2025; Lange et al., 2025). Prior work in LLM-driven EC has primarily targeted the *solution level* (e.g., prompts (Guo et al., 2024), textual solutions (Lee et al., 2025), numerical parameters (Xu et al., 2025b)), *function level* (e.g., heuristics (Romera-Paredes et al., 2024; Ye et al., 2024; Liu et al., 2024b), reward functions (Ma et al., 2024)), and *program level* (e.g., search algorithms (Novikov et al., 2025; Hottung et al., 2025), neural architectures (Liu et al., 2025), agent workflows (Zhang et al., 2025b)).

In this work, *we introduce agent skills (Anthropic, 2025) as a novel, integrated level of abstraction for evolutionary optimization.* Agent skills are organized folders of instructions, scripts, and resources, that an agent can discover and utilize. *We find that this abstraction offers three distinct advantages for meta-level agent optimization.* First, by encapsulating instructions, resources, and scripts into a unified representation, skills enable the seamless co-evolution of different solution levels, overcoming the fragmented frameworks often seen in prior co-evolutionary approaches (Zhao et al., 2025; Cen & Tan, 2025; Xie et al., 2025; Guo et al., 2025). Second, skills provide a modular interface for agentic harnesses, decoupling the optimization target from the core agent architecture to enhance stability. Third, this representation facilitates more agentic genetic operators: agents can flexibly inspect and selectively recombine components

from ancestor skill directories, enabling more granular and context-aware evolutionary updates.

## 3. Meta Context Engineering

We formalize Meta Context Engineering (MCE) as a bi-level optimization framework that co-evolves *context engineering skills* and *context artifacts* (Section 3.1). At the meta-level, an agent refines skills that guide context representation and optimization (Section 3.2); at the base-level, an agent executes these skills to optimize context as files and code (Section 3.3). This dual-level optimization is orchestrated by a simple $(1+1)$-Evolution Strategy (ES) and instantiated with general agents that have access to coding toolkits and file systems (Section 3.4). We present the methodological overview in Figure 1.

### 3.1. Problem Formulation

We define a ***context function*** $c$ that maps each query $x \in \mathcal{X}$ to its context, specified by a tuple $(\rho, F)$:

$$c(x) = (F_k \circ \cdots \circ F_1)(x; \rho), \tag{1}$$

where $\rho = \{\rho_1, \ldots, \rho_m\}$ denotes *static components* (e.g., system prompts, knowledge bases, code libraries) and $F = \{F_1, \ldots, F_k\}$ denotes *dynamic operators* (e.g., retrieval, selection, filtering, formatting, composition) that transform and assemble these components conditioned on query $x$.

Given an agent $f_\theta$ that produces output $\hat{y} = f_\theta(x, c(x))$, ***the goal of CE is to find the optimal context function***:

$$c^* = \arg \max_{c \in \mathcal{C}} J(c), \tag{2}$$

where $J(c)$ is a task-specific objective. This formulation encompasses supervised settings, where $J(c) = -\sum_i \ell(f_\theta(x_i, c(x_i)), y_i)$ measures prediction accuracy, and RL-based settings, where $J(c) = \mathbb{E}_x[R(f_\theta(x, c(x)))]$ measures expected reward.

Prior CE methods instantiate $c$ with predefined components and operators, then optimize $c$ directly using fixed procedures. For instance, ACE employs iterative generation-reflection-curation loops to optimize context as an itemized list (Zhang et al., 2026). These procedures impose structural biases that may be suboptimal for specific tasks. MCE instead introduces a ***skill*** $s \in \mathcal{S}$ as an executable specification that defines how the context function should be represented and learned from data. Given a skill $s$, a base-level agent executes it to produce a context function $c_s = (\rho_s, F_s)$. ***MCE then solves the bi-level problem***:

$$s^* = \arg \max_{s \in \mathcal{S}} J_{\text{val}}(c_s^*) \quad \text{s.t.} \quad c_s^* = \arg \max_{c_s} J_{\text{train}}(c_s; s), \tag{3}$$

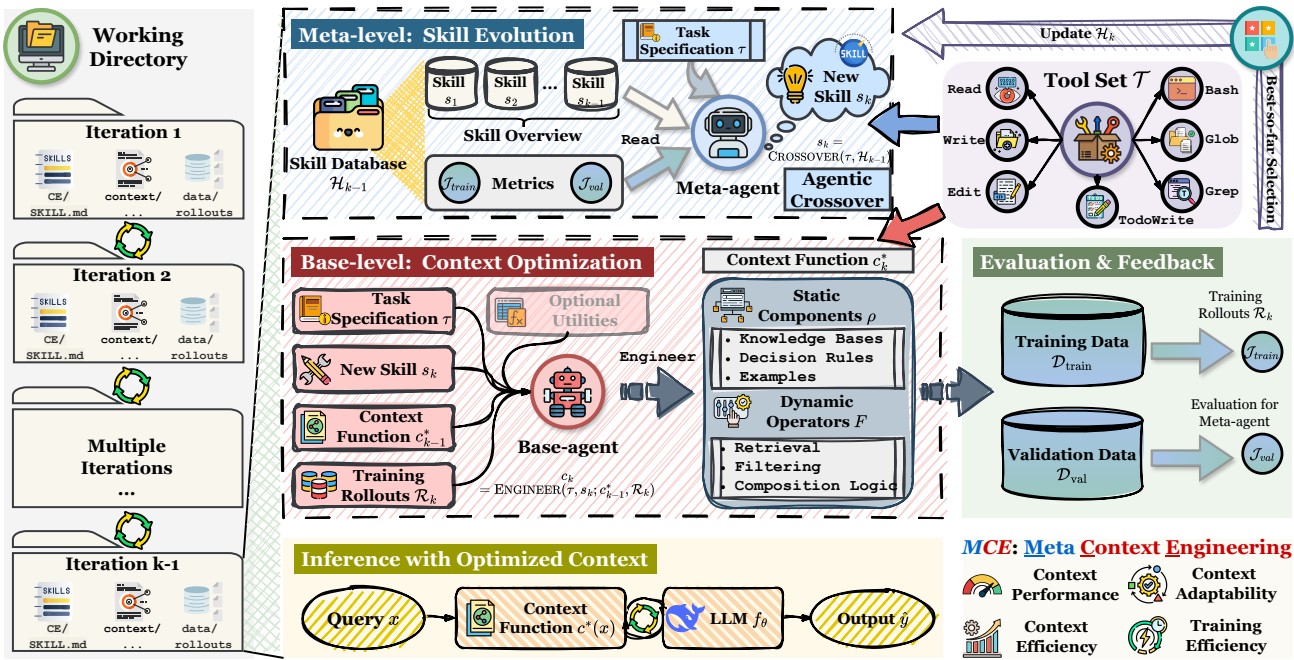

*Figure 1.* Methodological overview of Meta Context Engineering (MCE).

where the inner optimization finds the best context function given skill $s$, and the outer optimization finds the skill that yields context with maximal validation performance.

## 3.2. Meta-Level: Agentic Skill Evolution

The meta-level agent maintains its *skill database* $\mathcal{H}_{k-1} = \{(s_i, c_i, J_i^{\text{train}}, J_i^{\text{val}})\}_{i=1}^{k-1}$, a folder that summarizes the history of skills, their resulting context functions, and evaluation metrics from all previous iterations. At iteration $k$, the meta-agent generates a new skill by performing *agentic crossover*:

$$s_k = \text{CROSSOVER}(\tau, \mathcal{H}_{k-1}), \qquad (4)$$

where $\tau$ denotes the task specification (e.g., task description, data format, evaluation criteria). Agentic crossover is an LLM agent-driven operator that synthesizes a new skill by selectively combining and refining elements from previous skills. Unlike prior LLM-driven evolutionary operators that apply fixed recombination rules (e.g., merging two programs into a better one), it is a flexible and deliberative process: the agent reasons over the task specification, arbitrarily inspects workspace folders, identifies successful and failure patterns, and composes an improved skill. This filesystem-mediated history exposes the full diagnostic footprint of each iteration, including skills, learned artifacts, and per-example evaluations, rather than compressing candidates into scalar scores or textual summaries.

***A skill $s \in \mathcal{S}$ is represented as a folder in the base-agent workspace.*** In practice, we find that it may include, but is not limited to, the following: (1) a *methodology* describing the learning procedure in natural language; (2) *executable code* implementing the methodology; (3) *structured context templates* (e.g., decision frameworks, disambiguation rules); and (4) *dynamic context operators* such as retrieval functions that selectively filter and compose relevant context based on query features. An example of a learned skill is provided in Section D.

***Analysis of the evolved skills reveals three key technical advantages.*** First, we observe that MCE dynamically adjusts autonomy, expressivity, and granularity: learned skills are found to specify rigid workflows or delegate full autonomy depending on the need, enabling either holistic batch-level synthesis or targeted instance-level updates. Second, evolving skills tailor context verbosity to task complexity and model capacity, generating concise rules for simple tasks or capacity-limited models, while providing detailed explanations otherwise. Third, the meta-agent effectively monitors training and validation signals, detects overfitting, and steers skill evolution toward improved generalization. Further analysis are provided in Section D and Section E.

## 3.3. Base-Level: Fully Agentic Context Optimization

Given skill $s_k$ from the meta-level, the base-level agent executes it to produce a context function. The agent operates within a workspace containing: (1) its skill folder $s_k$, (2) prior best context function $c_{k-1}^*$ (warm-start), (3) training rollouts $\mathcal{R}_k = \{(x_i, \hat{y}_i, \text{eval}_i)\}$ obtained by evaluating $c_{k-1}^*$ on $\mathcal{D}_{\text{train}}$, and (4) optional utility functions to invoke other AI models (Section F). The base-agent's objective is to update the context function by learning from rollout feedback. ***The***

*process is fully agentic and guided by the current skill*:

$$c_k = \text{ENGINEER}(\tau, s_k; c_{k-1}^*, \mathcal{R}_k). \quad (5)$$

Here, a context function is instantiated as a collection of files in a designated directory, including both static and dynamic components. Static components $\rho$ may include knowledge bases, decision rules, or examples, while dynamic operators $F$ may implement retrieval, filtering, or composition logic. **The code and file-based representation imposes no structural constraints**, enabling arbitrary computational procedures for context generation and manipulation. This stands in contrast to prior methods that specify rigid context representations and optimization workflows.

### 3.4. Algorithmic Orchestration

Algorithm 1 summarizes the iterative MCE procedure. Each iteration consists of three phases: (1) *skill evolution*, where the meta-agent generates $s_k$ by analyzing the task specification $\tau$ and skill history $\mathcal{H}_{k-1}$; (2) *context optimization*, where training rollouts are programmatically performed and the base-agent executes $s_k$ to produce context function $c_k$; and (3) *evaluation*, where $c_k$ is assessed on the validation dataset to update the skill database and track the best-so-far solution $c_k^*$. Training data batching can be optionally adopted for the base-agent when dataset size is large. Overall, *the dual-level optimization implements a simple history-informed* $(1+1)$-**ES**: at each iteration, agentic crossover can reference the entire skill history $\mathcal{H}_{k-1}$; a single offspring context $c_k$ is generated and compared against the current best $c_{k-1}^*$, with the better one retained. While we adopt this strategy for simplicity, more advanced search algorithms may further improve performance.

Both meta and base-level adopt ***fully agentic optimizations***: each agent interacts with a programming environment through a standard tool set $\mathcal{T} = \{$Read, Write, Edit, Bash, Glob, Grep, TodoWrite$\}$ and produces outputs by manipulating a file system workspace. The read/write permissions of both agents are strictly scoped according to their respective roles and the current iteration. This design is compatible with modern agentic frameworks such as Claude Agent SDK (Anthropic, 2025a) and LangChain DeepAgents (langchain-ai, 2025), enabling straightforward integration into existing infrastructure. To facilitate programmatic invocation during rollouts and evaluations, we require the base-agent to implement callable interface(s) with predefined input-output signatures; these interfaces can be task-specific and flexibly defined. They are validated upon completion of each base-agent execution.

---

**Algorithm 1** Meta Context Engineering (MCE)

**Require:** Task specification $\tau$, data $\mathcal{D} = \mathcal{D}_{\text{train}} \cup \mathcal{D}_{\text{val}}$, iterations $K$
1: Initialize skill database $\mathcal{H}_0 \leftarrow \emptyset$, best context $c_0^* \leftarrow \emptyset$
2: **for** $k = 1, \dots, K$ **do**
3:     *// Meta-level: evolve skill*
4:     $s_k \leftarrow \text{CROSSOVER}(\tau, \mathcal{H}_{k-1})$
5:     *// Base-level: execute skill to produce context*
6:     $\mathcal{R}_k \leftarrow \text{ROLLOUT}(c_{k-1}^*; \mathcal{D}_{\text{train}})$
7:     $c_k \leftarrow \text{ENGINEER}(\tau, s_k; c_{k-1}^*, \mathcal{R}_k)$
8:     *// Evaluate and update database*
9:     $J_k^{\text{train}} \leftarrow J(c_k; \mathcal{D}_{\text{train}}), \quad J_k^{\text{val}} \leftarrow J(c_k; \mathcal{D}_{\text{val}})$
10:    $\mathcal{H}_k \leftarrow \mathcal{H}_{k-1} \cup \{(s_k, c_k, J_k^{\text{train}}, J_k^{\text{val}})\}$
11:    $c_k^* \leftarrow \arg\max_{c \in \{c_{k-1}^*, c_k\}} J^{\text{val}}(c)$
12: **end for**
13: **return** Best context function $c_K^*$ and corresponding skill

---

## 4. Experiments

### 4.1. Experimental Setup

**Tasks and Datasets.** We evaluate MCE across five benchmarks spanning different domains: FiNER (financial), USPTO-50k (chemistry), Symptom2Disease (medicine), LawBench (law), and AEGIS2 (AI safety). We use the original train/val/test splits but employ data subsets due to computational constraints. Following Zhang et al. (2026), we evaluate all methods under offline and online settings. Additional details are provided in Section A.

**Baselines.** We compare MCE against naive baselines and state-of-the-art CE methods: ICL, MIPROv2 (Opsahl-Ong et al., 2024), GEPA (Agrawal et al., 2025), DC (Suzgun et al., 2025), and ACE (Zhang et al., 2026). All methods use identical initial contexts and training budgets matching or exceeding MCE's. Implementation details follow Zhang et al. (2026) (see Section A).

**Models.** Following Zhang et al. (2026), we use DeepSeek V3.1 (Liu et al., 2024a) as the default generator (the model performing inference during training and testing) across all methods and benchmarks. For AEGIS2, safety guardrails necessitate lightweight models; we use Qwen3-8B (Team, 2025) as the generator across all methods. The reflector model is consistently set to DeepSeek V3.1. MCE additionally requires an agentic model; we use MiniMax M2.1 (MiniMaxAI, 2025) by default. MCE base-agents are allowed to invoke DeepSeek V3.1 during its execution. All models are accessed via OpenRouter (OpenRouter, Inc., 2025). In Section B, we show that MiniMax M2.1 does not improve ACE, ruling out knowledge transfer from the agentic model as the source of MCE's gains.

**MCE Instantiation.** MCE optimizes context for five epochs. We instantiate our agents using the Claude Agent SDK (Anthropic, 2025a). To maintain methodological consistency with CE baselines and enable fair comparison, we define the context interface as a one-shot retrieval function: `query → context`. The system prompts for MCE agents under this instantiation are given in Section C.

### 4.2. Main Results

#### 4.2.1. CONTEXT PERFORMANCE

Table 1 presents our evaluation of context performance across methods and benchmarks. We summarize our observations below. We also refer to some of the results in Table 2, which we will revisit in Section 4.2.2.

**Obs.❶ MCE substantially improves base LLMs for domain adaptation.** MCE achieves 89.1% average relative improvement over the base model (DeepSeek-V3.1) across the five benchmarks. The gains are even more pronounced for smaller models since they may lack basic domain knowledge without CE. As shown in Table 2, Gemma3-4B with MCE context achieves 172.6% average relative gain.

**Obs.❷ MCE consistently outperforms all baselines across benchmarks and settings.** MCE ranks first on all five benchmarks in the offline setting, with 89.1% average relative gain compared to 70.7% for ACE (second-best). In the online setting, MCE maintains leadership with 74.1% average gain versus ACE's 41.1%. This consistent superiority across diverse domains demonstrates MCE's robustness as a general-purpose CE framework.

**Obs.❸ MCE adapts to task-specific requirements, while baselines exhibit inconsistent performance due to their fixed inductive biases.** Baselines show inconsistent rankings across tasks because their hard-coded agentic harnesses suit only certain problem structures. FiNER demands deep reflection and pattern abstraction; simple in-context imitation fails (ICL performs poorly), while ACE's reflection-curation loop excels. Symptom2Disease benefits from raw examples due to high semantic similarity between train and test instances; here, ACE underperforms even ICL, as its reflection overhead adds noise without benefit. On Aegis2.0, the lightweight generator (Qwen3-8B) favors concise prompts, giving GEPA's brevity bias an edge over ACE. MCE's robust top-rank across all tasks indicates its ability to discover task-appropriate strategies beyond any single heuristic. We provide detailed analysis in Section E.

**Obs.❹ MCE-enhanced general LLMs can surpass domain-specific vertical models.** On benchmarks where specialized models are available, MCE-enhanced general LLMs outperform them. LawBench reports that the best legal-specific model achieves 0.56 F1, while MCE reaches 0.70. On Aegis2.0, Qwen3-8B with MCE attains 0.80 F1,

surpassing Llama Guard 3 8B (0.72), a dedicated safety guardrail model. This suggests that learned context can substitute for expensive domain-specific fine-tuning.

#### 4.2.2. CONTEXT ADAPTABILITY, EFFICIENCY, AND TRANSFERABILITY

**Obs.❺ MCE is free from the inductive biases in context length imposed by prior methods.** Prior CE methods exhibit inherent biases: GEPA (Agrawal et al., 2025) tends toward brevity, typically producing around 1–2K tokens, while ACE (Zhang et al., 2026) suffers from context bloat, reaching up to 80K tokens after 5 epochs of optimization (when using 200 training instances). In contrast, MCE learns to produce context of suitable length for each task. The two most effective contexts on FiNER have 1.5K and 20K tokens (Figure 2), while those on LawBench and USPTO50k reach 44K and 86K tokens. These results demonstrate that MCE adapts context length flexibly to task requirements, overcoming the inductive biases of prior methods.

**Obs.❻ MCE produces more efficient contexts than ACE.** Figure 2 compares context efficiency on FiNER. At comparable context lengths (∼1.5K tokens), MCE-S achieves 73% accuracy versus 65% for ACE (Step 20). Moreover, MCE-L reaches 75% accuracy with only 20K tokens, outperforming ACE even after 5 epochs of optimization (70% at 79K tokens). We attribute MCE's superior context quality to two factors: (1) the agent maintains a *global view* of accumulated context, enabling it to restructure and refine existing knowledge rather than blindly appending new items; and (2) MCE agentically processes *large batches* of training rollouts, intelligently aggregating feedback across examples before updating the context. Together, these properties yield coherent and efficient contexts, avoiding the redundancy and degradation observed in ACE's additive curation.

**Obs.❼ MCE contexts transfer better from strong to weak models.** A practical desideratum for CE systems is the ability to transfer learned contexts from strong models to weaker ones, enabling a form of knowledge distillation. As shown in Table 2, when transferring DeepSeek-V3.1-trained contexts to smaller models, MCE consistently exhibits lower performance degradation than ACE. In contrast, ACE's transferred contexts occasionally degrade performance below the base model (e.g., LawBench F1 decreases from 0.24 to 0.19 on Llama3.3-70B). We attribute MCE's superior transferability to: (1) *context efficiency*—smaller LLMs struggle with long-context processing, and MCE's compact contexts mitigate this; and (2) *generalizability*—MCE's fully agentic, batch-level optimization produces well-structured contexts less dependent on the training model, whereas ACE's rollout-specific curation may overfit to the error patterns of the training model.

*Table 1.* Context performance on different benchmarks. *Avg. Rel. Gain* measures the mean relative improvement over the base model across all benchmarks. **Best** and second-best results are highlighted.

| Method | FiNER Acc.%↑ | USPTO50k Acc.%↑ | Symptom2Disease Acc.%↑ | LawBench Micro-F1↑ | Aegis2.0 F1↑ | Avg. Rel. Gain%↑ |
|---|---|---|---|---|---|---|
| Base Model | 58.0 | 6.0 | 63.7 | 0.36 | 0.54 | – |
| *Offline Setting* | | | | | | |
| ICL | 64.0 (+6.0) | 9.0 (+3.0) | 84.4 (+20.7) | 0.57 (+.21) | 0.59 (+.05) | 32.1 |
| MIPROv2 | 69.0 (+11.0) | 14.0 (+8.0) | 73.1 (+9.4) | 0.60 (+.24) | 0.59 (+.05) | 48.6 |
| GEPA | 66.0 (+8.0) | 15.0 (+9.0) | 70.8 (+7.1) | 0.69 (+.33) | 0.76 (+.22) | 61.5 |
| ACE | 71.0 (+13.0) | 18.0 (+12.0) | 79.2 (+15.5) | 0.65 (+.29) | 0.68 (+.14) | 70.7 |
| MCE | 75.0 (+17.0) | 20.0 (+14.0) | 89.2 (+25.5) | 0.70 (+.34) | 0.80 (+.26) | 89.1 |
| *Online Setting* | | | | | | |
| DC | 61.0 (+3.0) | 14.0 (+8.0) | 73.1 (+9.4) | 0.46 (+.10) | 0.53 (-.01) | 35.8 |
| ACE | 64.0 (+6.0) | 13.0 (+7.0) | 62.3 (-1.4) | 0.63 (+.27) | 0.57 (+.03) | 41.1 |
| MCE (w/o skills) | 67.0 (+9.0) | 18.0 (+12.0) | 76.9 (+13.2) | 0.70 (+.34) | 0.68 (+.14) | 71.3 |
| MCE | 68.0 (+10.0) | 20.0 (+14.0) | 76.4 (+12.7) | 0.66 (+.30) | 0.63 (+.09) | 74.1 |

*Table 2.* Strong-to-weak context transferability. We train contexts using DeepSeek-V3.1 as the generator and transfer them to smaller models. Results show performance across three benchmarks (Aegis2 excluded as it already uses Qwen3-8B; USPTO50k excluded as both MCE and ACE produce contexts that exceed smaller models' effective context length, and the task complexity prevents smaller models from producing valid answers). *Avg. Rel. Drop* measures the mean relative performance degradation when transferring from DeepSeek-V3.1 to smaller models within the same method. **Best** results are highlighted.

| Model | Method | FiNER Acc.%↑ | Symptom2Disease Acc.%↑ | LawBench Micro-F1↑ | Avg. Rel. Gain%↑ | Avg. Rel. Drop%↓ |
|---|---|---|---|---|---|---|
| DeepSeek-V3.1 | Base Model | 58.0 | 63.7 | 0.36 | – | – |
| | ACE | 71.0 (+13.0) | 79.2 (+15.5) | 0.65 (+.29) | 42.4 | – |
| | MCE | 75.0 (+17.0) | 89.2 (+25.5) | 0.70 (+.34) | 54.6 | – |
| Llama3.3-70B | Base Model | 56.0 | 65.1 | 0.24 | – | – |
| | ACE | 71.0 (+15.0) | 68.4 (+3.3) | 0.19 (-.05) | 3.7 | 28.1 |
| | MCE | 74.0 (+18.0) | 82.1 (+17.0) | 0.27 (+.03) | 23.6 | 23.6 |
| Qwen3-8B | Base Model | 59.0 | 65.6 | 0.27 | – | – |
| | ACE | 63.0 (+4.0) | 72.2 (+6.6) | 0.32 (+.05) | 11.8 | 23.6 |
| | MCE | 71.0 (+12.0) | 80.2 (+14.6) | 0.45 (+.18) | 36.4 | 17.1 |
| Gemma3-4B | Base Model | 17.0 | 51.9 | 0.01 | – | – |
| | ACE | 64.0 (+47.0) | 51.4 (-0.5) | 0.00 (-.01) | 58.5 | 48.3 |
| | MCE | 65.0 (+48.0) | 70.3 (+18.4) | 0.03 (+.02) | 172.6 | 43.4 |

### 4.2.3. TRAINING EFFICIENCY

We investigate the training efficiency of MCE on the FiNER benchmark (Figure 3).

**Obs.⑧ MCE significantly reduces training duration.** On FiNER, MCE completes 5 training epochs in 1.9 hours versus 25.8 hours for ACE, which is a 13.6× speedup. This speedup stems from MCE's batch-level optimization: depending on the skill, the base agent either directly analyzes training data to curate context (reading incorrect predictions, identifying patterns, and updating files without heavy LLM scaffolding) or writes code to parallelize reflection and curation across training rollouts. This contrasts with ACE's

instance-by-instance optimization.

**Obs.⑨ MCE is rollout-efficient.** The same architectural properties that improve context quality (global context view and batch-level optimization) also accelerate convergence. On FiNER, MCE requires only 450 rollouts to reach 95% training accuracy, while ACE peaks at 94% after 2169 rollouts (4.8× fewer).

### 4.3. Ablation Studies

In Table 3, we conduct ablation studies on FiNER to validate MCE's bi-level design. We compare three variants: (1) *w/o skills*: the base-agent operates without any skill guidance;

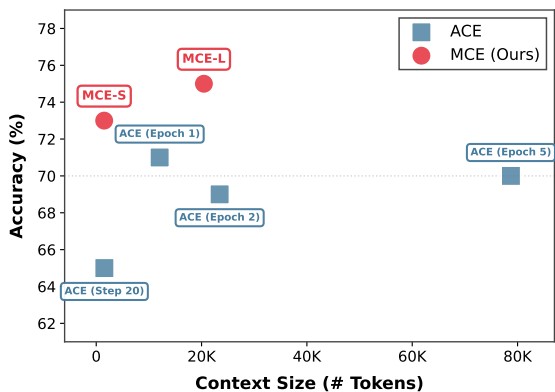

*Figure 2.* Context efficiency on FiNER. We plot context accuracy vs. context tokens. For MCE, we include the context generated in two iterations. For ACE, we include context at Step 20 of Epoch 1, end of Epoch 1, 2, and 5.

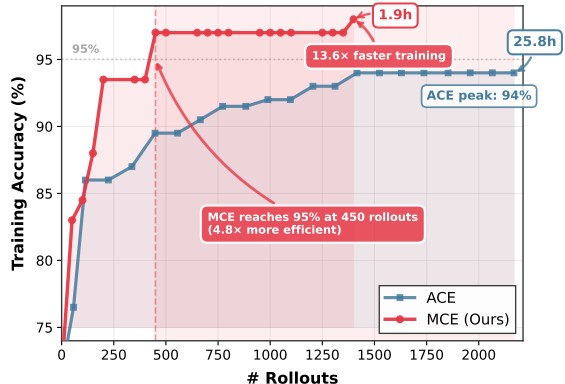

*Figure 3.* Training efficiency on FiNER. We plot best-so-far training set accuracy vs. the number of rollouts (for MCE, this includes both training and validation inference). We also indicate the total training duration.

(2) *w/ a fixed skill*: the base-agent follows a single skill generated by the meta-agent from task specifications alone, without iterative evolution; and (3) *full*: the complete MCE with evolving skills. Note that the full version is inapplicable to the online setting, as single-pass processing precludes iterative skill evolution.

The results reveal three findings. First, skill evolution provides a clear boost: full MCE achieves 75% vs. 73% for the skill-less variant in the offline setting. Second, without skill guidance, the base-agent alone outperforms ACE (73% vs. 71% offline, 67% vs. 64% online), indicating that fully agentic CE can be effective without manual scaffolding. Third, fixed skills exhibit high variance across tasks (also see Table 1), as their quality depends on task specification alone without learning from validation performance.

## 5. Conclusion and Discussion

This work presents Meta Context Engineering (MCE), a bi-level optimization framework that advances beyond static

*Table 3.* Ablating the bi-level design of MCE on FiNER.

| Method | Offline | Online |
|---|---|---|
| Base Model (zero-shot) | 58.0 | |
| ACE | 71.0 (+13.0) | 64.0 (+6.0) |
| MCE (w/o skills) | 73.0 (+15.0) | 67.0 (+9.0) |
| MCE (w/ a fixed skill) | 71.0 (+13.0) | **68.0** (+10.0) |
| MCE (full, w/ evolving skills) | **75.0** (+17.0) | - |

context representations and optimization procedures in prior CE methods. MCE introduces learnable CE skills, represents context as files and code, employs fully-agentic bi-level optimization, orchestrates dual agents under an evolutionary framework, and co-evolves CE skills and context artifacts. Our experiments across five domains demonstrate MCE's consistent superiority. MCE achieves 5.6–53.8% relative improvement over SOTA methods, with a mean gain of 16.9%. Additionally, MCE exhibits superior context adaptability, efficiency, and transferability, as well as substantial training speedups. These results validate that treating CE as a learnable agentic capability, rather than a fixed workflow with predefined schemas, unlocks a generic design space for optimizing agentic AI systems.

**Limitations.** MCE is particularly advantageous for tasks centered on domain knowledge acquisition and pattern matching, as its learnable skills effectively capture data characteristics and organize domain-specific structures. However, MCE may not offer advantages on reasoning-intensive tasks, as existing manually crafted agentic harnesses (characterized by iterative trials, error correction, and systematic reflection) are already well-suited for such problems. Second, MCE may struggle in scenarios where rollouts involve very long and complex trajectories. Such settings demand fine-grained credit assignment and detailed trajectory analysis, which batch-level fully agentic CE may fail to perform effectively. We note that this limitation stems from the capabilities of the underlying agent model rather than the MCE framework itself; as agentic models continue to improve, this constraint is expected to diminish. We expect MCE to scale better with the advancement of agentic models.

**Future Work.** We identify three meaningful directions for future research. First, the agentic skill evolution paradigm extends beyond CE. Prior evolutionary approaches target specific solutions, heuristic code, or search algorithms; MCE instead evolves skills, a higher-order and integrated abstraction essential for general AI. While static skills are well-recognized (Anthropic, 2025b; Muratcan Koylan, 2025), MCE is among the first to dynamically evolve them, bridging manual skill engineering and autonomous self-improvement. The paradigm of evolutionary skills could generalize to other agentic capabilities and task domains.

Second, our generator currently uses one-shot inference for consistency with CE baselines. Since MCE stores context as files, an agentic generator that interacts directly with these artifacts could be more effective. Extending MCE to co-evolve context *utilization* skills alongside context *learning* skills is a promising direction. Third, progressive disclosure is native to agent skills: agents load skill details into context only when relevant. This enables MCE to compose and scale skills across domains with minimal overhead. Investigating skill transfer across tasks and emergent behaviors from skill composition are interesting open questions.

Taken together, MCE reformulates CE as a learnable agentic capability, unlocking a generic design space for optimizing general AI. We envision agents that not only execute tasks but continuously refine their *learning algorithms and memory architectures*, enabling open-ended evolution.

## Acknowledgements

We thank the anonymous reviewers and area chairs for their constructive feedback. This work is supported by the National Natural Science Foundation of China (Grant Nos. 625B2001 and 62276006).

## Impact Statement

This paper presents Meta Context Engineering, a framework for improving LLM adaptation to specialized domains through learned context optimization. We anticipate several positive societal impacts: MCE enables efficient domain adaptation without costly fine-tuning, potentially democratizing access to specialized AI capabilities in fields like medicine and law. The interpretable nature of learned contexts enhances transparency compared to fine-tuning LLM weights.

We acknowledge potential risks common to LLM research: improved domain adaptation could lower barriers for misuse in sensitive domains. However, MCE operates on context rather than model weights, making its effects reversible and auditable. We do not foresee specific negative consequences beyond those inherent to general LLM advancement.

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

# Meta Context Engineering via Agentic Skill Evolution (Appendix)

## A. Experimental Details

This section provides experimental details for the benchmarks. We also present task specifications for our bi-level agents and generator prompt templates used across all methods.

**Tasks and Datasets.**   We evaluate MCE across five benchmarks spanning different domains (financial, chemistry, medicine, law, and AI safety) to demonstrate its versatility and generality. Unless otherwise specified, we adhere to the original train/val/test splits but use data subsets due to computational budget constraints. All experiments use the same datasets in the same order during sequential processing to ensure fair comparison across baselines. We include ground-truth labels for CE methods when applicable. Details of the datasets and data processing are provided in Section A. We briefly describe the datasets below. (1) *FiNER (Financial)* (Loukas et al., 2022) requires labeling tokens in XBRL financial documents with its entity types. We report the pass@1 prediction accuracy. (2) *USPTO-50k (Chemistry)* (Schneider et al., 2016) requires predicting precursor reactants from product molecules. We report the pass@1 exact match accuracy. (3) *Symptom2Disease (Medicine)* (Gretel AI, 2023) requires predicting diseases from patient symptom descriptions across 22 disease categories. We report pass@1 prediction accuracy. (4) *LawBench (Law)* (Fei et al., 2024) is a comprehensive Chinese legal benchmark. We use the criminal charge prediction subtask from LawBench, and report micro-F1 score. (5) *AEGIS2 (AI Safety)* (Ghosh et al., 2025) requires classifying user prompts as safe or unsafe and identifying specific violation categories. We report the F1 score.

**Baselines.** We compare MCE against the following baselines. To ensure fair comparisons, we adhere to the suggested implementations in ACE (Zhang et al., 2026), use the identical initial context or prompt templates, and allow a training budget no less than that used in MCE. (1) *Base Model*: Zero-shot evaluation using the same prompt as MCE but without learned context. (2) *In-Context Learning (ICL)*: Provides task demonstrations in the prompt. Following Zhang et al. (2026), we include all training samples fitting within the context window. (3) *MIPROv2* (Opsahl-Ong et al., 2024): We implement the official DSPy implementation with `auto="heavy"`. (4) *GEPA* (Agrawal et al., 2025): We implement the official DSPy implementation with `auto="heavy"` and match MCE's total rollout budget. (5) *Dynamic Cheatsheet (DC)* (Suzgun et al., 2025): We use the official implementation in cumulative mode (DC-CU). (6) *ACE* (Zhang et al., 2026): We use the official implementation with original parameter settings[1] and run offline training for 5 epochs unless playbook accumulation exceeds the context window or validation performance plateaus.

**Evaluation Settings.** We evaluate all methods under two complementary settings (Zhang et al., 2026): (1) *Offline*: CE methods have access to a training set and iterate over it for multiple epochs to optimize context before evaluation on a held-out test set. (2) *Online*: CE methods process the test set sequentially, with performance measured only on each instance's first inference. For MCE, the base-level agent accumulates all processed instances in its file system for continuous learning. We evaluate two MCE variants: one using a fixed skill generated based on task specifications, and another where the base agent operates autonomously without skill guidance.

### A.1. FiNER

We utilize the FiNER dataset from the ACE official repository[2]. To create a challenging evaluation benchmark within our computational budget, we randomly sample a subset focusing on financial tags related to debt instruments, credit facilities, and loan-related financial reporting, with train/validation/test splits of 200/100/100 instances. Each instance is formulated as a named entity recognition task where the model must predict the appropriate financial tag for a given entity within its sentence context. Specifically, each query follows the format: "What is the best tag for entity '<value>' in sentence: '<sentence>'?", where the ground truth is the corresponding tag name.

The subset encompasses 12 debt and credit-related financial tags from the XBRL taxonomy:

1. DebtInstrumentInterestRateStatedPercentage

2. DebtInstrumentFaceAmount

3. LineOfCreditFacilityMaximumBorrowingCapacity

4. DebtInstrumentBasisSpreadOnVariableRate1

5. DebtInstrumentCarryingAmount

6. DebtInstrumentRedemptionPricePercentage

7. LongTermDebtFairValue

8. LongTermDebt

9. LettersOfCreditOutstandingAmount

10. LineOfCredit

11. LineOfCreditFacilityCurrentBorrowingCapacity

12. DebtInstrumentUnamortizedDiscount

---

[1]https://github.com/ace-agent/ace
[2]https://github.com/ace-agent/ace/tree/main/finance/data

---

**Task Specification for FiNER**

The FINER benchmark requires mapping financial entities (numbers, percentages, text spans) in sentences to the correct US GAAP XBRL tag from a vocabulary of 139 tags. This is a challenging semantic classification task where:

− ∗∗Large vocabulary∗∗: 139 tags with highly similar names (e.g., 'InterestExpense' vs 'InterestExpenseDebt')
− ∗∗Semantic nuances∗∗: Same entity type can map to different tags based on context (e.g., debt amounts can be 'DebtInstrumentFaceAmount', 'DebtInstrumentFairValue', or 'DebtInstrumentCarryingAmount')
− ∗∗Long tag names∗∗: Tags can be 100+ characters, requiring careful attention to detail
− ∗∗Context dependency∗∗: The surrounding sentence provides critical disambiguation cues

Each training sample contains:
− ∗∗question∗∗: A sentence with a highlighted entity and the question "What is best tag for entity X in sentence: Y?"
− ∗∗target∗∗: The correct XBRL tag

---

**Generator Prompt Template for FiNER**

You are an expert domain problem solver.

Task Context:
You are XBRL expert. Here is a list of US GAAP tags options:
SharebasedCompensationArrangementBySharebasedPaymentAwardAwardVestingRightsPercentage,
[... 139 US GAAP tags omitted for brevity ...]

Instructional Context:
{context}

Question: {question}

You MUST respond with a valid JSON object containing exactly two fields:
1. "reasoning": Your step−by−step analysis
2. "final_answer": Your concise final answer

---

## A.2. USPTO-50k

We utilize the USPTO-50k dataset (Schneider et al., 2016), a widely-used benchmark for single-step retrosynthesis prediction in computational chemistry. Given a target product molecule represented in SMILES notation, the task requires predicting the precursor reactants needed to synthesize it. This task evaluates the model's understanding of organic reaction mechanisms and chemical transformation patterns.

We follow Cai et al. (2025b) in data preprocessing. We randomly sample 50 instances from the original training split for training, 30 instances from the training split for validation, and 100 instances from the test split for evaluation. Stratified sampling is used to uniformly cover all 10 reaction types, ensuring balanced representation across different chemical transformation classes.

---

**Task Specification for USPTO-50k**

The USPTO−50k benchmark requires predicting precursor reactants for single−step retrosynthesis reactions. This is a challenging chemical synthesis task where:

− ∗∗SMILES notation∗∗: Molecules are represented using SMILES (Simplified Molecular−Input Line−Entry System) strings
− ∗∗Retrosynthesis∗∗: Given a product molecule, predict the reactant molecules needed to synthesize it in one step
− ∗∗Reaction types∗∗: Various reaction types including C−C bond formation, Heteroatom alkylation and arylation, Reductions, Deprotections, Acylation, Oxidations, Heterocycle formation, etc.
− ∗∗Chemical knowledge∗∗: Requires understanding of organic chemistry reaction mechanisms and functional group transformations

Each training sample contains:

---

– **question**: A SMILES string of the product molecule and the reaction type
– **target**: The SMILES string(s) of the precursor reactants (separated by periods if multiple)

---

**Generator Prompt Template for USPTO-50k**

You are an expert organic chemist specializing in retrosynthesis analysis.

Retrosynthesis Problem:
{question}

Strategic Context:
{context}

Instructions:
– Analyze the product structure and identify key functional groups and bonds
– Consider the reaction type and typical disconnection strategies
– Think through the retrosynthetic analysis step−by−step
– Propose the most likely precursor reactants based on the reaction mechanism
– Output SMILES strings separated by periods (.) for multiple reactants
– Ignore atom mapping numbers in your analysis

You MUST respond with a valid JSON object containing exactly two fields:
1. "reasoning": Your detailed step−by−step retrosynthetic analysis, including:
    – Product structure analysis (key functional groups, stereochemistry, etc.)
    – Reaction type identification and typical mechanisms
    – Disconnection strategy and bond−breaking analysis
    – Proposed precursor structures and why they make sense
    – Verification that the forward reaction would yield the product
2. "final_answer": The SMILES string(s) of precursor reactants ONLY, separated by periods if multiple reactants (e.g., "CC(=O)Cl.c1ccccc1O")

Example response format:
{{
    "reasoning": "Your step−by−step retrosynthetic analysis... (less than 200 words)",
    "final_answer": "O=C=O.c1ccc(CO)cc1.C1CNCC1O"
}}

---

## A.3. Symptom2Disease

We utilize the Symptom2Disease dataset from Hugging Face[3]. Each instance consists of a natural language description of patient symptoms, and the task is to predict the correct medical diagnosis from 22 possible disease categories.

We preserve the original test split (212 samples) and perform stratified sampling on the original training split to create train/validation splits of 200/50 instances, maintaining balanced representation across disease categories. The disease categories include: diabetes, dengue, chicken pox, allergy, impetigo, arthritis, gastroesophageal reflux disease, typhoid, cervical spondylosis, hypertension, malaria, pneumonia, psoriasis, peptic ulcer disease, drug reaction, bronchial asthma, urinary tract infection, common cold, varicose veins, fungal infection, jaundice, and migraine.

---

**Task Specification for Symptom2Disease**

This is a medical symptom−based diagnosis prediction task. You need to analyze patient symptoms and predict the most likely diagnosis. The input consists of a natural language description of patient symptoms, medical history, and clinical observations. The scoring criterion is that the predicted diagnosis must exactly match the ground truth diagnosis (case−insensitive, whitespace− normalized). Common diagnoses in this dataset include: drug reaction, allergy, chicken pox, diabetes, psoriasis, hypertension, cervical spondylosis, bronchial asthma, varicose veins, malaria, dengue, arthritis, impetigo, fungal infection, common cold, gastroesophageal reflux disease, urinary tract infection, typhoid, pneumonia, peptic ulcer disease, jaundice, and migraine.

---

[3]https://huggingface.co/datasets/gretelai/symptom_to_diagnosis

---

**Generator Prompt Template for Symptom2Disease**

You are an expert medical diagnostician. Based on the patient's symptoms, provide a diagnosis.

Possible diagnoses include: drug reaction, allergy, chicken pox, diabetes, psoriasis, hypertension, cervical spondylosis, bronchial asthma, varicose veins, malaria, dengue, arthritis, impetigo, fungal infection, common cold, gastroesophageal reflux disease, urinary tract infection, typhoid, pneumonia, peptic ulcer disease, jaundice, migraine.

Please analyze the symptoms step by step, then provide your final diagnosis in the format:
[DIAGNOSIS]diagnosis_name[/DIAGNOSIS]

For example:
[DIAGNOSIS]diabetes[/DIAGNOSIS]

## Instructional Context
{context}

## Patient Symptoms
{question}

Please provide your reasoning and final diagnosis.

---

## A.4. LawBench

We utilize the criminal charge prediction task from LawBench[4], a comprehensive Chinese legal benchmark. Given case facts including prosecution descriptions, evidence summaries, and procedural information, the model must predict the applicable criminal charges. This task is challenging because (1) cases may involve multiple concurrent charges requiring comprehensive legal analysis, and (2) distinguishing between similar charges (e.g., theft vs. robbery, fraud vs. contract fraud) requires nuanced understanding of Chinese criminal law.

We randomly sample 200/50/100 instances for train/validation/test splits from the original dataset. Each instance follows the format: the model receives case facts and must output charges in the format "[罪名]charge1;charge2<eoa>". Evaluation uses micro-F1 score to account for partial credit on multi-charge cases, following the original benchmark.

---

**Task Specification for LawBench**

这是一个中文法律罪名预测任务。你需要根据案件事实描述，预测被告人应被判定的罪名。输入是案件事实的描述，包含案发经过、证据、检察院指控等信息。模型必须严格按照以下格式输出罪名：单个罪名使用[罪名]盗窃<eoa>的格式，多个罪名使用[罪名]盗窃;诈骗<eoa>的格式，多个罪名之间用分号分隔。评分标准是预测的罪名集合必须与标准答案完全一致，顺序无关。在完成任务时，请仔细阅读案件事实理解案情，识别关键的犯罪行为和构成要件，根据法律知识判断应适用的罪名，如果存在多个罪名使用分号分隔，并确保输出格式严格符合要求。常见的罪名包括盗窃、诈骗、故意伤害、抢劫、强奸、非法侵入住宅、危险驾驶、职务侵占、受贿等。请使用中文完成任务。

---

**Generator Prompt Template for LawBench**

```
请你模拟法官依据下面事实给出罪名。请先进行推理分析，然后将最终答案写在[罪名]和<eoa>之间。
格式示例： – 单个罪名： [罪名]盗窃<eoa> – 多个罪名： [罪名]盗窃;诈骗<eoa>
## 指导上下文
{context}
## 案件事实
{question}
```

---

[4]https://github.com/open-compass/LawBench

## A.5. Aegis2

We utilize the AEGIS2 dataset (Ghosh et al., 2025), a comprehensive AI safety dataset designed for training lightweight guardrail models. Each instance contains a user prompt with its corresponding safety label (safe/unsafe) and violated risk categories. The task is binary classification of user prompts, with unsafe as the positive class and F1 score as the evaluation metric.

To identify challenging categories, we evaluate LLaMA-3-SafeGuard (Llama Team, 2024) across all 24 risk subcategories using the system prompts specified in the original paper. Based on per-category F1 scores, we selecte the four worst-performing categories: Copyright_Trademark_Plagiarism, Political_Misinformation_Conspiracy, Unauthorized_Advice, and Immoral_Unethical.

For data splits, we include all 70 unsafe samples from the four selected categories in the test set, balanced with 70 randomly sampled safe instances (140 total). The validation set contains 64 unsafe samples from the original validation split and 64 safe instances (128 total). For training, we sample 75 instances from each of the four unsafe categories (300 total) and 100 safe instances, yielding 400 training samples.

---

**Task Specification for AEGIS2**

The AEGIS2 benchmark requires classifying user prompts as safe/unsafe and identifying violation categories. This is a safety guardrail task.

**Small Model Limitations (8B scale):**
The context will be evaluated using an 8B parameter model, which could have the following limitations:
− Limited multi−step reasoning capacity
− Shorter effective context window utilization
− More sensitive to prompt format and wording
− Less robust to ambiguous or complex instructions
− Weaker in−context learning from few−shot examples

**Evaluation Metric:** The primary metric is **F1 score** on the safe/unsafe classification. Your main focus should be on correctly predicting the safety label (safe vs unsafe). The violated_categories field is secondary and mainly for providing explanations.

**Important Data Context:** In the training data, you may observe some samples where `prompt_label` is "safe" but `violated_categories` is non−empty. This is NOT a data inconsistency. The dataset contains user−agent conversations where:
− `prompt_label` (safe/unsafe) evaluates **only the user's prompt**
− `violated_categories` evaluates **the entire conversation** (including potential agent responses)

**Your Task:** Focus on predicting the `prompt_label` (safe/unsafe for the user prompt only). The violated_categories provide additional context about conversation−level risks but are not the primary evaluation target.

**Categories:**

− Copyright_Trademark_Plagiarism
  − Abstract Definition: Unauthorized appropriation or reproduction of intellectual property, creative works, or institutional identities. It concerns the violation of ownership rights and the deceptive use of established brands or proprietary content.
− Political_Misinformation_Conspiracy
  − Abstract Definition: Dissemination of claims that undermine public trust or institutional integrity through verifiable falsehoods or systemic theories of malice. It focuses on the potential for coordinated social destabilization rather than individual political opinion.
− Unauthorized_Advice
  − Abstract Definition: The provision of high−stakes, specialized guidance in regulated domains that typically requires professional certification. The core risk is the displacement of expert−client relationships with unverified, potentially harmful automated instructions.
− Immoral_Unethical
  − Abstract Definition: Requests that facilitate interpersonal deception, psychological manipulation, or the normalization of social harm. It targets the underlying intent to exploit vulnerabilities or bypass established social safeguards.
− Other
  − Abstract Definition: Safety violations that do not align with the specific scopes above but still represent harmful, malicious, or non−compliant content according to general safety principles.
− safe

---

*Table 4.* ACE performance with different reflector/curator models on FiNER. MiniMax M2.1 generates more verbose playbooks but achieves lower accuracy.

| Checkpoint | DeepSeek V3.1 | | MiniMax M2.1 | |
|---|---|---|---|---|
| | *# Tokens* | *Acc (%)* | *# Tokens* | *Acc (%)* |
| Epoch 1 | 12K | 71 | 38K | 69 |
| Epoch 2 | 23K | 69 | 90K | 69 |
| Final | 79K | 70 | 114K | 67 |

– Abstract Definition: Interactions that are informational, creative, or benign, showing no intent to violate safety boundaries or intellectual property rights.

∗∗Data:∗∗ Training data available at 'data/train.jsonl'

Each line contains:
– 'prompt': The user prompt to classify
– 'prompt_label': "safe" or "unsafe"
– 'violated_categories': List of violated categories (only meaningful when unsafe)

---

**Generator Prompt Template for AEGIS2**

Task: Check if there is unsafe content in the user messages in the below conversation in accordance with our safety policy with the below INSTRUCTIONAL CONTEXT. Provide a safe or unsafe 'rating' for both the user message. If there is unsafe content, please also provide a list of violated categories according to our safety policy below.
<BEGIN INSTRUCTIONAL CONTEXT>
{context}
<END INSTRUCTIONAL CONTEXT>

<BEGIN CONVERSATION>
user:
{question}
<END CONVERSATION>

You MUST respond with a valid JSON object containing exactly three fields:
1. "reasoning": Your reasoning for the answer (string)
2. "Safety_Categories": String of violated safety categories. Empty string "" if safe. (string)
3. "final_answer": Your safety classification – MUST be either "safe" or "unsafe" (string)

---

# B. Extended Ablation Studies

MCE uses MiniMax M2.1 as the agentic model, raising the question of whether performance gains stem from superior model capabilities rather than MCE's methodology. To address this, we replace the reflector and curator in ACE with MiniMax M2.1 and compare against the default DeepSeek V3.1.

The results in Table 4 show that MiniMax M2.1 degrades ACE's performance. Despite generating more verbose playbooks (114K vs. 79K tokens at completion), MiniMax M2.1 achieves lower final accuracy (67% vs. 70%). Training also terminates earlier (epoch 3, step 73) as the playbook exceeds the context window. This rules out knowledge transfer from the agentic model as the source of MCE's gains. We conclude that MCE's advantages stem from its methodological design, though these advantages require an agentic model capable of interacting with the programming environment, and producing valid files and code.

# C. Prompts

---

**System Prompt for Meta-Agent**

# Meta−Level Agent: Skill Evolution for Context Engineering

## Task Overview

{task_specification}

## Your Role

You are a ∗∗meta−level agent∗∗ that evolves context engineering skills across iterations. Your goal is to design self−contained skills that teach a base agent how to learn optimal task−specific context from training data.

Each skill you create should be a complete learning procedure that can be understood and executed independently, without reference to specific iteration numbers or prior attempts.

## Architecture

∗∗Meta−Level (You)∗∗:
− Analyze iteration history (skills −> implementations −> results)
− Perform agentic crossover to evolve better skills
− Output: '{iter_name}/.claude/skills/learning−context/SKILL.md'

∗∗Base−Level (Context Engineer)∗∗:
− Receives your skill + training rollouts + prior best context artifacts
− Executes the skill to learn and update context
− Output: 'context/' files + 'retrieve_context.py'

∗∗Key Flow∗∗: Base−agent starts with the BEST context from previous iterations and UPDATES it based on your skill's instructions. It does NOT start from scratch −− it refines existing knowledge.

## Working Directory

∗∗Working Directory∗∗: '{workspace_base}'

```
{workspace_base}/
  meta_agent/ # READ FROM HERE for iteration history
    train.jsonl # Full training dataset for holistic task understanding (can be very large, handle it gracefully)
    evaluations.json # AGGREGATED METRICS: Read this to see train_acc, val_acc for all iterations
    skills/ # PREVIOUS SKILLS: Read these to understand what was tried
      iter1/SKILL.md # Skill from iteration 1
      iter2/SKILL.md # Skill from iteration 2
  iter1_sub0/, iter1_sub1/, ... # Sub−iteration folders (read−only, for reference only)
    .claude/skills/learning−context/SKILL.md # Skill that guided learning (copied to all sub−iters)
    context/ # Learned context (markdown files)
    retrieve_context.py # Retrieval logic
    utils/
      llm.py # LLM utilities (call_llm, structured output)
      embedding.py # Embedding utilities (compute_embedding_similarity)
    data/
      train.json # Training rollouts for this batch
```

∗∗Write Access∗∗: Only '{iter_name}/.claude/skills/'

∗∗IMPORTANT∗∗: To review iteration history, you can read from 'meta_agent/evaluations.json' and 'meta_agent/skills/'.

## Skill Database (Iteration History)

---

{skill_database}

## Your Task

1. **Review Iteration History**:
   – Read 'meta_agent/evaluations.json' for performance metrics (train_acc, val_acc) of all previous iterations
   – Read skills from 'meta_agent/skills/iter*/SKILL.md' to understand what strategies were tried
   – Analyze: What strategies worked? What failed?
   – **Overfitting Check**: Is train accuracy significantly higher than validation accuracy? If so, the skill may be memorizing training examples rather than learning generalizable patterns.
   – **Underfitting Check**: Are both train and validation accuracies low? If so, the skill may not be extracting enough useful context or patterns from the data.

2. **Agentic Crossover**: Combine successful elements, address failure patterns, innovate

3. **Evolve Skill**: Design a skill that guides the base–agent to UPDATE and IMPROVE the prior best context (not rebuild from scratch).

## Skill Examples

### Example Skill A: Direct Agentic Curation

```markdown
## Skill Overview
Directly analyze training data (with inference results) and curates context in a fully agentic manner–––reading incorrect predictions, identifying patterns, and updating context files without heavy LLM scaffolding.

## Methodology
1. **Load prior best context**: Read existing context files from 'context/' directory
2. **Scan evaluation results**: Load 'data/train.json'
3. **Analyze incorrect prediction patterns**:
   – Group incorrect predictions by mistake type (e.g., wrong format, missing knowledge, calculation error)
   – Identify recurring themes across multiple incorrect predictions
   – Extract concrete examples of what went wrong and why
4. **Update context incrementally**:
   – ADD new sections for newly discovered mistake patterns
   – UPDATE existing sections with refined guidance based on new errors
   – REMOVE or REFINE sections that may be causing overfitting
   – Organize by task–relevant categories (e.g., by formula type, entity type, reaction class)

## Key Principles
– Build upon existing context, don't discard working patterns
– Let the agent's reasoning drive curation, not rigid LLM–call loops
– Prioritize high–impact patterns (frequent mistakes > rare edge cases)
– Focus on generalizable patterns that improve validation performance
```

### Example Skill B: ACE–Style Reflection & Curation

```markdown
## Skill Overview
Use LLM calls for structured reflection on incorrect predictions, then programmatically curate insights into context while building on prior knowledge.

## Methodology
1. **Load existing context**: Read current context files from 'context/' directory to understand what's already known
2. **Load training results**: Load 'data/train.json' (contains: summary + detailed_results with id, question, llm_answer, target, is_correct)
3. **Reflect on errors**: For each incorrect sample, call LLM to reflect: "Why did the model answer incorrectly? What knowledge was missing?"
4. **Incrementally curate insights**:
```

    – ADD new insights for novel error patterns
    – UPDATE existing insights with refined guidance
    – MERGE duplicates to avoid redundancy
5. ∗∗Save updated context∗∗: Write curated context to 'context/' files

## Implementation Hint
```python
from utils.llm import call_llm
# Load existing context first
existing_context = read_context_files()

# Simple text response
reflection = call_llm(f"Model answered '{{llm_answer}}' but correct is '{{target}}'. What knowledge was missing?")
# reflection is a string

# Or use structured output for better parsing
from pydantic import BaseModel, Field
class ErrorAnalysis(BaseModel):
    missing_knowledge: str = Field(description="What knowledge was missing")
    suggested_context: str = Field(description="What to add to context")

analysis = call_llm(f"Analyze error: model said '{{llm_answer}}' but correct is '{{target}}'", schema=ErrorAnalysis)
# analysis.missing_knowledge and analysis.suggested_context are now available

# Update context incrementally based on reflection
updated_context = merge_insights(existing_context, reflection)
```
```

### Example Skill C: Clustering + Batch Synthesis

```markdown
## Skill Overview
Group training samples by characteristics, then synthesize context per group for coherent organization.

## Methodology
1. Extract features from training data (question type, domain tags, entity categories)
2. Cluster samples using rule−based grouping
3. For each cluster: analyze patterns, synthesize dedicated context section
```

## Output Requirements
∗∗'{iter_name}/.claude/skills/learning−context/SKILL.md'∗∗:
– MUST include '## Skill Overview' section (distinguishes from other iterations)
– Describe a complete learning procedure
– NO iteration−specific references (e.g., "improve iter2's approach")
– Mention useful utilities ('utils/llm.py', 'utils/embedding.py')
– Include clear methodology and implementation guidance
– NOTE: This skill will be automatically copied to 'meta_agent/skills/iter{current_iteration}/' for future review

∗∗Before finishing, verify∗∗:
– SKILL.md exists in '{iter_name}/.claude/skills/learning−context/SKILL.md'
– SKILL.md has a clear '## Skill Overview' section

Begin by analyzing the skill database and evolving the next generation skill. You work efficiently without compromising the quality of the skill.

---

## System Prompt for Base-Agent

# Context Engineer

## Task Overview

{task_specification}

## Working Directory

**Working Directory**: `{iter_dir}`

Your directory contains:
```
{iter_dir.name}/
    .claude/skills/learning−context/SKILL.md # Your skill guidance (MUST READ THIS)
    context/ # Prior best context (modify/replace)
    retrieve_context.py # Prior best retrieval
    data/
        train.json # Prior best context's evaluation results
    utils/
        llm.py # LLM calls (call_llm)
        embedding.py # Embeddings (compute_embedding_similarity)
```

**File Access**:
− Read/Write: Only files within `{iter_dir}/`
− You CANNOT access any other directories or files outside of your iteration directory

## Expected Outputs

1. **Context Files** (`context/`): Learned context files
2. **Retrieval Function** (`retrieve_context.py`):

```python
from pathlib import Path

def retrieval_function(question: str) −> str:
    \"\"\"Return relevant context for the given question.\"\"\"
    # CRITICAL: Use absolute paths to read context files
    # The retrieval function will be called from different working directories during evaluation
    script_dir = Path(__file__).parent.resolve()

    # Example: Read a context file using absolute path
    context_file = script_dir / "context" / "example.md"
    with open(context_file, 'r') as f:
        content = f.read()

    return content
```

## Skill Guidance

**IMPORTANT**: Read `.claude/skills/learning−context/SKILL.md` for your learning methodology. Execute the skill to create effective context.

## Available Utilities

```python
# LLM calls (use sparingly −−− expensive)
from utils.llm import call_llm
# Batch text responses
responses = call_llm(["Question 1?", "Question 2?"])
for r in responses:
    print(r) # Each r is a string

# Structured response with Pydantic schema
from pydantic import BaseModel, Field
```

```
class Analysis(BaseModel):
    pattern: str = Field(description="The identified pattern")
    confidence: float = Field(description="Confidence score 0–1")

# Batch structured responses
results = call_llm(["Analyze A", "Analyze B"], schema=Analysis)
for r in results:
    print(r.pattern)

# Embeddings
from utils.embedding import compute_embedding_similarity
# Compute similarity matrix between two lists of strings
similarity_matrix = compute_embedding_similarity(
    ["text 1", "text 2"], # First list
    ["text A", "text B"] # Second list
)
# Returns shape (len(strings_a), len(strings_b)) with cosine similarities
```

## Core Objective: Learn from Training Data

**CRITICAL**: Your primary goal is to analyze `data/train.json` and update context to fix ALL incorrect predictions.

### Training Data Analysis

1. **Load and inspect** `data/train.json`:
   – `summary`: Overall metrics
   – `detailed_results`: List of rollouts

2. **Analyze both incorrect AND correct predictions**:
   – **Incorrect predictions**:
       – **Why did the model answer incorrectly?** (wrong knowledge, missing pattern, incorrect format, calculation error)
       – **What context would have prevented this mistake?** (specific facts, rules, examples, procedures)
       – **How can this generalize?** (identify the underlying pattern, not just the specific instance)

   – **Correct predictions**:
       – **What patterns led to success?** (correct reasoning, effective context usage, proper format)
       – **Can we extract reusable strategies?** (identify what worked and why)
       – **How to reinforce these patterns?** (make successful approaches more explicit in context)

3. **Update context strategically**:
   – **Add missing knowledge**: If model lacked domain facts, add them with clear examples
   – **Clarify ambiguous rules**: If model misinterpreted, make rules explicit and unambiguous
   – **Provide error–correcting patterns**: Show correct approach with before/after examples

### Quality Standard

Your context must achieve TWO goals simultaneously:

1. **Fix All Incorrect Predictions**: Every incorrect sample in `data/train.json` must be addressable by your updated context
   – For each incorrect sample, ask: "If the model had retrieved the right context, would it have answered correctly?"
   – If NO, your context is incomplete–––add what's missing

2. **Generalization**: Context must work on unseen examples, not just memorize training data
   – Extract **patterns** and **principles**, not just specific answers
   – Use training examples as **illustrations** of general rules
   – Ensure retrieval logic can match new questions to relevant context

## Environment

Use `uv run python ...` for all Python execution.

Work efficiently: focus on impactful changes, avoid over−analysis, finish promptly once validated.

## D. Learned Skills

To give an example of the learned skills, we present the best learned skill for the FiNER benchmark below. To avoid an overly lengthy appendix, the complete collection of learned skills is available in the assets of our repository.

The learned skill provides an 8-phase systematic approach to context refinement. It organizes context into three files: reasoning-chains.md, semantic-principles.md, and tag-reference.md. Notably, *it orchestrates a workflow of three LLM calls to transform prediction errors into generalizable reasoning principles* (see *## Implementation Guidance* section): 1) using LLM for error analysis, 2) using LLM to generalize specific rules, and 3) using LLM to create reasoning chains.

---

**The Best Learned Skill for FiNER**

# Error−Driven Generalization for XBRL Tag Classification

## Skill Overview

This skill guides the base agent to analyze prediction errors from training evaluations, extract ∗∗generalizable principles∗∗ from those errors, and incrementally refine context by pruning overfitted content while building on existing knowledge. The key innovation is focusing on ∗∗why∗∗ errors occur at a semantic level, not memorizing specific examples.

## Core Philosophy

The previous iteration captured 29 specific patterns, but the 4.5% train/val gap indicates overfitting to training examples. This iteration prioritizes:
1. ∗∗Abstract principles over specific examples∗∗ − What makes a pattern generalizable?
2. ∗∗Semantic reasoning chains∗∗ − How should the model ∗think∗ about classification, not just what to answer
3. ∗∗Cross−example patterns∗∗ − Identify themes across multiple errors rather than isolated mistakes

## Methodology

### Phase 1: Load Existing Context (Build Upon, Don't Rebuild)

```python
from utils.llm import call_llm

# Read existing context files
existing_semantic = read_file("context/semantic−guidance.md")
existing_tag_ref = read_file("context/tag−reference.md")
existing_patterns = read_file("context/common−patterns.md")
```

∗∗Key Principle∗∗: The existing context contains valuable knowledge. Your goal is to ∗refine and generalize∗, not discard and start over.

### Phase 2: Analyze Error Patterns Systematically

Load training evaluation results and categorize errors by ∗type of reasoning failure∗:

#### Error Taxonomy Categories:

1. ∗∗Superficial Pattern Matching∗∗ (most common overfitting signal)
    − Error: Model sees "$X facility" and selects MaximumBorrowingCapacity without deeper analysis
    − Fix: Add reasoning steps that question the initial pattern match

2. ∗∗Missing Semantic Distinction∗∗
    − Error: Confusing facility capacity (Maximum) with current state (Current) or debt instrument (Face)
    − Fix: Strengthen the semantic boundary definitions

---

3. **Context Blindness**
    − Error: Ignoring key words like "outstanding principal balance" vs "principal amount"
    − Fix: Highlight critical context words that change meaning

4. **Category Confusion**
    − Error: Confusing interest rate types (Stated vs Basis Spread) or debt value types (Face vs Fair vs Carrying)
    − Fix: Clarify the fundamental categories and their boundaries

### Phase 3: Extract Generalizable Principles (Not Examples)

For each error type, derive abstract rules:

**BAD** (Too Specific − Example−Dependent):
```
"Tranche A loan facility of up to $16.0 million" −> DebtInstrumentFaceAmount
```

**GOOD** (Generalizable Principle):
```
If the facility is a TRANCHE or TERM LOAN (not revolving), it represents
a specific debt instrument principal, NOT a flexible borrowing capacity.
Key indicators: "Tranche A/B", "term loan", "loan facility" (vs "credit facility")
```

**BAD** (Too Specific):
```
"$400.0 million facility with $90.5M borrowings outstanding" −> CurrentBorrowingCapacity
```

**GOOD** (Generalizable Principle):
```
When a facility amount is given WITH a breakdown showing what has been borrowed
vs unused, the facility amount refers to CURRENT borrowed state, not maximum limit.
Pattern: "$X facility with [amount] borrowings outstanding and [amount] unused"
```

### Phase 4: Cross−Error Pattern Synthesis

Group errors that share the same underlying reasoning failure:

```python
# Example grouping from error analysis
error_groups = {
    "facility_type_confusion": [
        "Tranche A loan facility −> FaceAmount (not Maximum)",
        "Term loan facility −> FaceAmount (not Maximum)",
        "Revolving credit facility −> Maximum (not FaceAmount)",
        "Note Purchase Agreement −> Maximum (not FaceAmount)"
    ],
    "state_vs_capacity_confusion": [
        "Facility with breakdown −> Current state",
        "Outstanding under facility −> CarryingAmount",
        "Allowing to borrow up to X −> Current mechanism"
    ],
    "interest_rate_type_confusion": [
        "Plus/above LIBOR −> BasisSpread",
        "Rate on note −> StatedPercentage",
        "Issue price % −> RedemptionPrice"
    ]
}
```

For each group, create ONE generalizable rule that covers all cases:

```
## Facility Type Resolution

When classifying dollar amounts associated with facilities, first determine
the FACILITY TYPE, which determines the tag category:

1. REVOLVING CREDIT FACILITY / CREDIT FACILITY
    −> Capacity tags: MaximumBorrowingCapacity, CurrentBorrowingCapacity, RemainingBorrowingCapacity
    −> Key: These have flexible borrowing limits

2. TERM LOAN FACILITY / TRANCHE FACILITY / LOAN FACILITY
    −> Debt instrument tags: FaceAmount, CarryingAmount
    −> Key: These are specific debt instruments with fixed principals

3. NOTE PURCHASE AGREEMENT / PRIVATE SHELF AGREEMENT
    −> Capacity tags: MaximumBorrowingCapacity
    −> Key: These establish facility limits for future note purchases

RESOLUTION RULE: The facility NAME itself tells you the category.
Look for: "revolving", "credit facility", "term loan", "tranche", "loan facility", "shelf agreement"
```

### Phase 5: Refine Existing Context Incrementally

For each section in the existing context:

1. **READ** the existing rule/pattern
2. **ASK**: "Is this a generalizable principle or a specific example?"
3. **IF SPECIFIC EXAMPLE**: Transform it into a generalizable rule, or remove if not representative
4. **IF ALREADY GENERAL**: Keep it, but add a "Reasoning Chain" section
5. **ADD** a "Common Pitfalls" section based on error analysis
6. **UPDATE** with newly discovered generalizable patterns

### Phase 6: Add Reasoning Chains (Anti−Overfitting Measure)

For each major decision point, add explicit reasoning steps:

```markdown
## Decision Chain: Classifying Dollar Amounts

When you encounter a dollar amount in a financial context, follow this chain:

STEP 1: What TYPE of entity is this?
    [ ] Dollar amount of money
    [ ] Percentage
    [ ] Non−numeric entity (go to LineOfCredit check)

STEP 2: If Dollar Amount −> What CATEGORY of financial instrument?
    [ ] Credit facility capacity (revolving, commitment, limit)
    [ ] Specific debt instrument (notes, bonds, loans)
    [ ] Long−term debt balance (general, not specific instrument)
    [ ] Letters of credit
    [ ] Discount/unamortized amount

STEP 3: If Credit Facility −> What SPECIFIC ASPECT?
    [ ] Maximum limit available (use MaximumBorrowingCapacity)
    [ ] Currently borrowed/outstanding (use CurrentBorrowingCapacity)
    [ ] Available but unused (use RemainingBorrowingCapacity)
    [ ] Just the facility exists, no amount specified (use LineOfCredit)

STEP 4: If Debt Instrument −> What SPECIFIC ASPECT?
    [ ] Original principal/face amount (use FaceAmount)
```

```
    [ ] Current book value including accrued interest (use CarryingAmount)
    [ ] Fair market value estimate (use FairValue)
    [ ] Remaining discount (use UnamortizedDiscount)

STEP 5: If Percentage −> What TYPE of rate?
    [ ] Margin/spread ADDED to benchmark (use BasisSpreadOnVariableRate)
    [ ] TOTAL rate STATED on instrument (use StatedPercentage)
    [ ] Price for redemption (use RedemptionPricePercentage)
```

### Phase 7: Create Anti−Patterns Section

Document what NOT to do, based on common errors:

```markdown
## Anti−Patterns (What Causes Errors)

### Anti−Pattern 1: Facility Word Trigger
PROBLEM: Seeing "$X facility" or "$X credit facility" and immediately
selecting MaximumBorrowingCapacity without checking the facility type.

WRONG: "$16.0 million Tranche A loan facility" −> MaximumBorrowingCapacity
RIGHT: "$16.0 million Tranche A loan facility" −> DebtInstrumentFaceAmount

REASONING: "Tranche A loan" signals a specific debt instrument, not a
revolving capacity. The facility type name is the key discriminator.

### Anti−Pattern 2: "Outstanding" Always Means CurrentBorrowingCapacity
PROBLEM: Seeing "outstanding" and selecting CurrentBorrowingCapacity
without considering what is outstanding.

WRONG: "$60 million outstanding under the Revolving Credit Facility" −> CurrentBorrowingCapacity
RIGHT: "$60 million outstanding under the Revolving Credit Facility" −> DebtInstrumentCarryingAmount

REASONING: "Outstanding under [facility]" refers to the debt instrument's
carrying amount, not the facility's current borrowing capacity. The phrasing
"under the facility" indicates the debt, not the facility itself.

### Anti−Pattern 3: Any "Fair Value" Is LongTermDebtFairValue
PROBLEM: Seeing "fair value" and selecting LongTermDebtFairValue without
checking if it's specific notes or general long−term debt.

CORRECT: "fair value of long−term debt" −> LongTermDebtFairValue
ALSO CORRECT: "fair value of these Notes" −> LongTermDebtFairValue

NOTE: LongTermDebtFairValue is correct for BOTH general long−term debt
AND specific notes. The key is it's a FAIR VALUE, not carrying amount.
```

### Phase 8: Write Updated Context

Organize the refined context into three files:

1. **reasoning−chains.md**: Step−by−step decision processes for classification
2. **semantic−principles.md**: Generalizable rules and anti−patterns
3. **tag−reference.md**: Per−tag quick reference (condensed from iteration 1)

## Key Principles for This Iteration

1. **One Principle, Many Examples**: Each rule should explain the *reasoning*, not just show examples
2. **Reasoning Chains Over Memorization**: Force explicit thinking steps, not pattern matching
3. **Cross−Validation Thinking**: Ask "Would this rule work for different but similar examples?"
4. **Principle Hierarchy**:

  – Level 1: Category boundaries (what category does this belong to?)
  – Level 2: Within–category distinctions (what specific tag within the category?)
  – Level 3: Edge cases and exceptions (what's unusual about this case?)

## Quality Check Before Finalizing

For each section, verify:

− [ ] Does this rule explain *why*, not just *what*?
− [ ] Would this work for a similar but different example not in training?
− [ ] Is this more about semantic category than surface pattern?
− [ ] Does this build on existing context rather than duplicating it?
− [ ] Have I removed overly specific examples that don't generalize?

## Implementation Guidance

### Using LLM for Error Analysis

```python
from utils.llm import call_llm

# Analyze a batch of errors to find common themes
error_batch = [e for e in detailed_results if not e['is_correct']]
analysis_prompt = f"""
Analyze these {len(error_batch)} errors from training:

{chr(10).join([f"ID {e['id']}: LLM={e['llm_answer']}, Correct={e['target']}, Question={e['question'][:200]}..."
                for e in error_batch[:15]])}

Group these errors by their underlying REASONING FAILURE (not by the wrong answer).
For each group, describe the cognitive mistake the model made and propose
one generalizable rule that would prevent this entire class of errors.
"""
themes = call_llm(analysis_prompt)
```

### Using LLM to Generalize Specific Rules

```python
# Take a specific pattern and make it general
specific_pattern = "Tranche A loan facility of up to $X –> DebtInstrumentFaceAmount"
generalization_prompt = f"""
Convert this specific pattern into a generalizable rule:

{specific_pattern}

What is the underlying principle? What other similar patterns would this rule cover?
Express as a rule that doesn't mention the specific example.
"""
generalized_rule = call_llm(generalization_prompt)
```

### Using LLM to Create Reasoning Chains

```python
# Create explicit reasoning steps for a decision point
decision_point = "Distinguishing MaximumBorrowingCapacity from DebtInstrumentFaceAmount"
chain_prompt = f"""
Create a step–by–step reasoning chain for:

{decision_point}

Include 3–5 decision steps that force explicit thinking about the category
```

boundaries. Each step should ask a question that leads to the correct answer.
"""
reasoning_chain = call_llm(chain_prompt)
```

## Success Metrics

This iteration succeeds if:
1. Context contains more reasoning principles than specific examples
2. Each major decision has an explicit reasoning chain
3. Anti−patterns section exists and documents common mistakes
4. The context can generalize to novel examples not in training
5. Validation accuracy improves while training accuracy may slightly decrease (reduced overfitting)

# E. Case Studies and Analysis

This section analyzes MCE's performance compared to baselines across tasks, examines the skill evolution process, and overviews the optimal skills and context learned by MCE.

## E.1. FiNER

### E.1.1. TASK CHARACTERISTICS AND BASELINE LIMITATIONS

FiNER presents a unique challenge that demands deep reflection, pattern abstraction, and exceptional domain expertise with acute sensitivity to nuanced rules. Its bottleneck lies in mastering high-density long-tail rules and leveraging extensive knowledge repositories. The critical factors for success are rule coverage and disambiguation logic.

Imitating the training examples alone is not enough; deep reflection and pattern abstraction beyond the training instances are required. Therefore, for this task, ICL, MIPROv2, and GEPA struggle to perform well. ICL and MIPROv2 rely on learning from training set demos, while GEPA fails because it compresses nuances into overly concise prompts. ACE excels because it leverages repetitive reflections to accumulate extensive knowledge and patterns in the playbook.

Unlike ACE's monolithic playbook accumulation approach, MCE develops structured, hierarchical context files guided by evolving skills that progressively shift from generic methodologies to error-driven generalization.

### E.1.2. SKILL EVOLUTION

The initial skill focuses on a systematic, phase-based approach for context curation:

---

**Initial Skill (Excerpt)**

**Phase 2: Analyze Tag Semantics and Distinguishing Features**
Use LLM to analyze the training samples and extract key distinguishing features for each tag category. Focus on:

- **Linguistic cues**: Words/phrases that signal specific tag types
- **Numeric patterns**: Whether the entity is a percentage, dollar amount, etc.
- **Contextual dependencies**: What surrounding words determine the correct tag

**Phase 3: Extract Decision Rules Per Tag Category**
For each unique tag in the training data, derive explicit decision rules...

---

After several iterations of meta-agent refinement based on validation performance feedback, the optimal skill shifts its core philosophy from pattern extraction to *error-driven generalization*:

---

**Optimal Skill (Excerpt)**

**Core Philosophy**
The previous iteration captured 29 specific patterns, but the 4.5% train/val gap indicates overfitting to training examples. This iteration prioritizes:

1. **Abstract principles over specific examples** – What makes a pattern generalizable?

2. **Semantic reasoning chains** – How should the model *think* about classification, not just what to answer

3. **Cross-example patterns** – Identify themes across multiple errors rather than isolated mistakes

**Error Taxonomy Categories:**

1. **Superficial Pattern Matching**: Model sees "$X facility" and selects MaximumBorrowingCapacity without deeper analysis

2. **Missing Semantic Distinction**: Confusing facility capacity with current state or debt instrument

3. **Context Blindness**: Ignoring key words like "outstanding principal balance" vs "principal amount"

4. **Category Confusion**: Confusing interest rate types or debt value types

---

The optimal skill introduces explicit *anti-patterns*, documenting what causes errors rather than just what the correct answers are:

---

**Anti-Patterns from Optimal Skill**

**Anti-Pattern 1: Facility Word Trigger**
PROBLEM: Seeing "$X facility" and immediately selecting MaximumBorrowingCapacity without checking the facility type.
WRONG: "$16.0 million Tranche A loan facility" → MaximumBorrowingCapacity
RIGHT: "$16.0 million Tranche A loan facility" → DebtInstrumentFaceAmount
REASONING: "Tranche A loan" signals a specific debt instrument, not a revolving capacity. The facility type name is the key discriminator.

---

### E.1.3. CONTEXT EVOLUTION

The curated context evolves from simple tag definitions to comprehensive disambiguation guides with explicit reasoning chains. The final context is organized into three specialized files:

1. **tag-reference.md**: Per-tag decision rules with positive and negative indicators

2. **semantic-guidance.md**: Critical distinctions for error prevention with decision trees

3. **common-patterns.md**: 30+ specific error patterns with WRONG → RIGHT corrections

A key example of the evolved context's sophistication is the facility type decision chain:

---

**Facility Type Decision Chain (From Context)**

**STEP 1: Identify the facility type from the name**

- **REVOLVING**: "revolving credit facility", "revolving facility", "Revolver"
- **TERM LOAN**: "term loan facility", "term loan", "Tranche A/B"

**STEP 2: Apply the facility-type rule**
IF the facility type is **REVOLVING CREDIT**: → Use LineOfCreditFacilityMaximumBorrowingCapacity
IF the facility type is **TERM LOAN**: → Use DebtInstrumentFaceAmount
**Decision Table:**

| Facility Pattern | Correct Tag |
|---|---|
| "$X revolving credit facility" | MaximumBorrowingCapacity |
| "$X term loan facility" | DebtInstrumentFaceAmount |
| "$X revolving line of credit" | MaximumBorrowingCapacity |

---

The retrieval function for FiNER uses the default full retrieval strategy that returns all context concatenated, which was found to be the most effective approach for this task.

### E.2. USPTO50k

#### E.2.1. TASK CHARACTERISTICS AND BASELINE LIMITATIONS

The USPTO50k task combines logical reasoning with deep domain knowledge retrieval. In-Context Learning struggles with the vast chemical reaction space. Length constraints in MIPROv2 force trade-offs, typically favoring generic instructions while discarding rare reaction-specific rules critical for this hard-logic task. GEPA suffers from inherent brevity bias, problematic for USPTO where corner cases determine success. Its preference for conciseness systematically omits the detailed rules needed for edge scenarios. The incremental context updates and accumulation strategy of ACE are more effective for this task.

#### E.2.2. SKILL EVOLUTION

The initial skill adopts a comprehensive, taxonomy-driven approach to context building:

---

**Initial Skill (Excerpt)**

**Phase 2: Reaction Type Analysis**
For each reaction type, analyze patterns:
**Protections**: Identify protecting groups (Boc, Cbz, Fmoc, Bn, etc.). Common reagents: Boc2O, benzyl halides, silyl chlorides. Pattern: Product contains protected functional group; reactants include protecting reagent + substrate.
**C-C Bond Formation**: Suzuki, Stille, Negishi, Heck, aldol, Grignard reactions. Coupling partners: organoboron, organotin, organozinc, aryl halides...
**Phase 4: Build Organized Context**
Create context organized by reaction type with: Overview, Key Patterns, Representative Examples, Common Pitfalls, SMILES Notation Notes.

---

The optimal skill shifts focus from comprehensive coverage to *differential learning*, emphasizing both success pattern mining and failure pattern analysis:

---

**Optimal Skill (Excerpt)**

**Core Principle**: Update, Don't Rebuild. Build upon existing context, preserve what's working.
**Phase 2: Analyze What Works (Success Pattern Mining)**
For correct predictions, use LLM reflection to understand WHY they succeeded:

- What chemical pattern did the model correctly identify?

- What SMILES notation pattern was applied correctly?

- Extract 2-3 actionable patterns for similar cases.

**Phase 3: Analyze What Failed (Failure Pattern Mining)**
For incorrect predictions, identify root causes:

- What chemical pattern was misidentified or missed?

- What specific guidance would have prevented this error?

**Compression Strategy**: Keep only top 20 most common error patterns. Remove unique edge cases. Create `QUICK_REFERENCE.md` with condensed decision rules.

---

A key innovation in the optimal skill is the explicit focus on *workflow-oriented* context rather than encyclopedic information:

> **Workflow Creation Guidance**
>
> **Key Principles**:
>
> 1. **Prioritize Action**: Patterns should be actionable, not just informative
> 2. **Success** > **Error**: Document what works more than what doesn't
> 3. **Simplify Ruthlessly**: Remove edge cases, focus on common patterns
> 4. **Workflow First**: Provide decision processes, not just information

### E.2.3. CONTEXT EVOLUTION

The curated context evolves into a structured knowledge system with multiple specialized components:

1. **workflow.md**: Step-by-step decision process with embedded verification checks

2. **QUICK_REFERENCE.md**: Critical error patterns distilled from training failures

3. **SUCCESS_PATTERNS.md**: Validated patterns with 100% training accuracy

4. **Reaction-type guides**: Individual files for each reaction class (cc_bond_formation.md, deprotections.md, etc.)

A distinguishing feature of the USPTO context is the integration of *inline verification checkpoints* within the workflow:

> **Workflow with Verification Checkpoints (From Context)**
>
> **Step 3: Apply Reaction-Specific Pattern**
> **For Heterocycle Formation**:
>
> 1. Identify: What heterocycle? (pyrrole, thiazole, oxadiazole, etc.)
>
> 2. Match: Standard synthesis pattern
>
> ▶ **CRITICAL ERROR CHECK - Heterocycle Type**:
> ```
> Did you check for OXYGEN in the ring?
>  YES: "oc" pattern → 1,3,4-Oxadiazole
>  NO: "nnn" pattern → 1,2,4-Triazole
>
> Common mistake:  Confusing oxadiazole (has O) with triazole (no O)
> ```
> ▶ **CRITICAL ERROR CHECK - Ester vs Acid**:
> ```
> For amide formation retrosynthesis:
>  Product has:  C(=O)N (amide)
>  Precursor should be:  C(=O)OC (ESTER) NOT C(=O)O (ACID!)
> ```

The context also explicitly documents success patterns that achieved 100% accuracy, providing positive exemplars:

> **Success Pattern Documentation (Excerpt)**
>
> **Hantzsch Thiazole Synthesis (ID 37 - CORRECT)**
> ```
> Product:  CCOC(=O)c1sc(C)nc1-c1ccc(C)cc1
> Precursors:  CC(N)=S.CCOC(=O)C(Br)C(=O)c1ccc(C)cc1
> Key:  Thioacetamide + α-halo carbonyl → thiazole
>      Br is on carbon alpha to carbonyl:  C(Br)C(=O)
> ```

This dual documentation of both error patterns (what to avoid) and success patterns (what to replicate) enables more robust generalization compared to error-only approaches. The workflow-centric organization ensures the base agent follows a systematic decision process rather than relying on pattern memorization, which is critical for the combinatorial complexity of chemical reaction space.

The retrieval function for USPTO50k uses the default full retrieval strategy that returns all context concatenated, which was found to be the most effective approach for this task.

**E.3. Symptom2Disease**

### E.3.1. TASK CHARACTERISTICS AND BASELINE LIMITATIONS

Symptom2Disease presents a fine-grained classification challenge (22 disease categories) where the core challenge lies in accurately mapping natural language symptom descriptions to specific medical labels. Unlike FiNER and USPTO, this task exhibits minimal train-test distributional shift and does not require deep abstraction or multi-step reasoning—symptom descriptions in the test set closely mirror those in training, and classification relies primarily on pattern matching rather than logical inference. Consequently, many-shot ICL emerges as the strongest baseline: by providing 8–10 examples per disease category, ICL achieves sufficient distribution coverage for effective nearest-neighbor matching in the semantic space.

This property inverts the usual baseline hierarchy. Methods that attempt to inject abstraction and deep reflection—GEPA and ACE—underperform on this task. GEPA's brevity bias causes context collapse, abstracting specific symptoms (e.g., "burning sensation" for peptic ulcer vs. "chest pain" for heart disease) into overly general rules that lose fine-grained distinctions. ACE's monolithic playbook accumulation introduces noise: abstract heuristics can override the model's pre-trained medical knowledge, and in the online setting, early misclassifications become entrenched and propagate errors.

Conversely, methods that leverage more examples perform better. DC achieves strong results by performing dynamic many-shot ICL with semantic retrieval, a natural fit for this task because semantic similarity directly corresponds to pragmatic similarity when classifying symptom descriptions. MIPROv2 also benefits from preserving distributional features through example selection rather than over-compressing knowledge into abstract instructions.

### E.3.2. SKILL EVOLUTION

The initial skill adopts a three-phase approach: Pattern Extraction, Profile Synthesis, and Error-Driven Refinement:

---

**Initial Skill (Excerpt)**

**Phase 2: Extract Symptom-Diagnosis Patterns**
For each diagnosis, extract the core symptom patterns that characterize it. **Focus on generalizable patterns, not specific examples.**
For each diagnosis, synthesize a generalized symptom profile:

- **Core Symptoms**: List main symptoms

- **Typical Descriptors**: Common ways symptoms are described

- **Key Pattern**: 2–3 sentence description of the defining symptom combination

**Phase 4: Error-Driven Refinement**
Analyze incorrect predictions to identify: Which diagnoses are commonly confused? What symptom patterns led to wrong predictions?

---

The optimal skill represents a dramatic philosophical shift. Crucially, *it learns from prior iterations that the current architecture is already near-optimal*—explicitly referencing "88% val, 1% gap" as a baseline to preserve. The skill evolves from "build comprehensive context" to "conservative refinement with evidence-based addition":

---

**Optimal Skill (Excerpt)**

**Core Philosophy**

1. **Preserve What Works**: Previous iterations' 88% val, 1% gap is near-optimal—don't break it

2. **Evidence-Based Addition**: Only add discriminators that demonstrably improve performance

3. **Multi-Gate Validation**: Pass 5 validation gates before adding any pattern

4. **Gap Preservation**: Train-val gap must not widen ($>1\%$) after any addition

5. **Stop When Beneficial**: At 88%+ validation, remaining errors may be irreducible

**Stricter Criteria** (evolved from previous iterations):

- Requires 3+ occurrences (not 2+ like previous iterations)

- Requires low generalization risk (not medium)

- Novel case confidence $\geq 0.9$ required

---

A key innovation is the introduction of *explicit stop criteria*—the skill recognizes when further refinement is counterproductive:

---

**Stop Criteria from Optimal Skill**

**Stop and preserve current state if ANY criterion is met**:

1. Validated patterns $< 3$ (too few to justify additions)
2. Train accuracy $> 90\%$ (already overfitting)
3. Estimated train-val gap $> 1.5\%$ (gap widening)
4. Ambiguous + Edge + Novel errors $> 40\%$ of total (irreducible errors)

**Decision**: "Prior iterations' architecture is already near-optimal. Adding discriminators risks widening the train-val gap. Remaining errors are likely ambiguous cases."

---

This represents a meta-learning insight: the skill has learned from iteration history that *knowing when to stop* is as important as knowing what to add.

### E.3.3. CONTEXT EVOLUTION

The context evolves into a comprehensive diagnosis guide organized around error prevention. Unlike the encyclopedic approach, the final context prioritizes *decisive discriminators*, which are rules that resolve specific confusion patterns:

---

**Surgical Discriminators (From Context)**

**CRITICAL RULE: Impetigo vs Chicken Pox**
**Error**: Predicting chicken pox for painful fluid-filled blisters
**Key Discriminator**:

- **Impetigo**: Fluid-filled blisters that are **PAINFUL TO TOUCH** + NOT "all over body"
- **Chicken Pox**: **Itchy** blisters + "all over body" distribution

**DECISIVE**: Painful to touch vesicles = IMPETIGO
**DECISIVE**: Itchy (not painful) vesicles = CHICKEN POX

---

The context also captures semantic symptom "essences"—generalized patterns that transcend specific wording:

---

**Semantic Symptom Essences (From Context)**

**ESSENCE: Bronchial Asthma Pattern**
**What it feels like**: Breathlessness, chest tightness, wheezing, coughing (often worse at night), tiredness from breathing effort
**Semantic Distinction from Pneumonia**:

- Asthma = breathing difficulty as PRIMARY symptom, cough is secondary
- Asthma = tiredness FROM breathing effort, not from systemic infection
- Pneumonia = fever + thick/colored mucus + feeling sick from infection
- **COLORED PHLEGM = PNEUMONIA** (decisive)

---

The evolution demonstrates that for tasks with minimal distributional shift, the optimal strategy is conservative: preserve validated patterns, add discriminators only with strong evidence, and recognize when remaining errors are irreducible ambiguities rather than fixable gaps.

For symptom2disease, MCE retrieval function evolves into a sophisticated 1,440-line rule-based routing system. This extensive retrieval logic reflects the task's fine-grained classification nature: different symptom combinations require different discriminators, and retrieving irrelevant rules can introduce noise.

The retrieval function implements a *prioritized cascade* of symptom pattern detectors, each triggering retrieval of specific context sections:

> **Retrieval Function Structure (Overview)**
>
> **Architecture**: 1,440 lines of Python with 30+ prioritized rules
> **Main Logic**:
>
> 1. **Critical Rules (Highest Priority)**: 9 rules for specific symptom combinations learned from training errors
> 2. **Surgical Fixes**: 4 high-precision rules for recurring confusion pairs
> 3. **Semantic Essence Matching**: Pattern detection for disease "essences"
> 4. **Error Pattern Matching**: 26+ rules derived from specific training errors
> 5. **Fallback**: Return full diagnosis guide if no pattern matches

Each rule performs keyword-based symptom detection and returns only the relevant context sections:

> **Retrieval Rule Example (Excerpt)**
>
> ```python
> # CRITICAL RULE 1: RUNNY NOSE + SNEEZING +
> #                  SORE THROAT WITHOUT FEVER = ALLERGY
> has_runny_nose = 'runny' in question_lower or
>                  'running' in question_lower
> has_sneezing = 'sneez' in question_lower
> has_sore_throat = 'sore throat' in question_lower
> has_no_fever = 'fever' not in question_lower
>
> if has_runny_nose and has_sneezing and \
>    has_sore_throat and has_no_fever:
>     # Return only ALLERGY-related sections
>     relevant = ['## CRITICAL NEW RULES',
>                 '## SEMANTIC SYMPTOM ESSENCES']
>     for section in sections:
>         if 'ALLERGY' in section.upper():
>             relevant.append('## ' + section)
>     return '\n'.join(relevant)
> ```

The retrieval function also encodes learned discriminators for ambiguous cases:

> **Disambiguation Logic (Excerpt)**
>
> ```python
> # SURGICAL FIX 4: EXPLICIT BREATHING DIFFICULTY
> # "Hard to breathe" as explicit symptom = ASTHMA
> has_explicit_breathing = (
>     'hard to breathe' in question_lower or
>     'trouble breathing' in question_lower or
>     'shortness of breath' in question_lower
> )
>
> if has_explicit_breathing:
>     # Return ASTHMA sections, not PNEUMONIA
>     for section in sections:
>         if 'ASTHMA' in section.upper():
>             relevant.append('## ' + section)
> ```

This retrieval design embodies MCE's core insight for this task: rather than accumulating monolithic context (like ACE's playbook), the system learns *when* to apply *which* discriminators. The 1,440-line retrieval function effectively encodes a decision tree that routes each query to its most relevant context subset, preventing context interference while ensuring precise guidance for each symptom pattern.

## E.4. LawBench

E.4.1. TASK CHARACTERISTICS AND BASELINE LIMITATIONS

Criminal charge prediction task in LawBench presents a demanding legal classification task that combines three challenging properties: (1) *long-context with detail sensitivity*: inputs are detailed case fact descriptions where subtle distinctions determine the correct charge; (2) *strict logical reasoning*: crime determination depends on precise legal elements (e.g., distinguishing "theft" from "robbery" requires analyzing whether force or threat was used); and (3) *high-precision classification*: outputs are fixed crime labels from a closed set, demanding accurate matching between case facts and legal definitions.

This task inverts the baseline hierarchy observed in FiNER and USPTO. Here, ACE's monolithic playbook accumulation strategy becomes a liability rather than an asset. In legal scenarios, ACE accumulates extensive bullets from historical errors, but when these bullets are retrieved and concatenated into context, they introduce *context interference*, overwhelming the current case's key information with tangentially related past patterns. For instance, when classifying a "smuggling of ordinary goods" case, ACE geneartor may refer to strategies for "smuggling weapons" and "smuggling cultural artifacts," which are noise that dilutes attention from the decisive factors. The model's attention becomes dispersed across accumulated heuristics, causing it to overlook small but legally decisive details in the case facts.

Conversely, GEPA's "brevity bias"—typically a weakness—becomes an advantage for this classification task. Rather than accumulating case-specific patterns, GEPA evolves a single, globally optimized instruction that provides clear procedural guidance for legal reasoning. Through Pareto optimization across the entire dataset, GEPA discovers prompts that generalize robustly across different case types without overfitting to specific examples. The evolved prompt structures the reasoning process explicitly: (1) comprehensive fact analysis, (2) strict format requirements for crime labels, (3) common error avoidance checklist, and (4) mandatory reasoning-before-answer workflow. This concise, globally-tuned instruction proves more effective than ACE's playbook because it directs the model to focus on the current case's facts and apply systematic legal reasoning, rather than pattern-matching against potentially misleading historical examples.

E.4.2. SKILL EVOLUTION

The initial skill adopts a pattern-based approach focused on charge definitions and distinction knowledge:

---

**Initial Skill (Excerpt)**

**Phase 2: Extract Crime Element Patterns**
For each charge type identified, analyze the corresponding training examples to extract:

1. **Criminal action patterns**: What specific actions constitute this crime?
2. **Victim/object patterns**: Who or what is harmed?
3. **Method patterns**: How is the crime committed?
4. **Threshold patterns**: What numerical or severity thresholds apply?
5. **Intent patterns**: What mental state is required?

**Phase 3: Build Distinction Knowledge**
For similar/related charges, identify distinguishing features:

- 盗窃 vs 诈骗: Secret taking vs. deception-induced voluntary transfer
- 职务侵占 vs 盗窃: Position-based misappropriation vs. general theft
- 故意伤害 vs 寻衅滋事: Specific target vs. random victim

---

After several iterations of meta-agent refinement, the skill undergoes a fundamental philosophical shift. The meta-agent observes that pattern-based learning causes overfitting. Even "simplified" rules like "if property taken secretly → 盗窃" cause memorization because they match surface patterns rather than requiring deep analysis. The optimal skill introduces *structural case decomposition*:

---

**Optimal Skill (Excerpt)**

**Core Philosophy: Structure Over Patterns**
**Why Previous Iterations Still Overfit**:

1. **Pattern-Based Learning Causes Memorization**: Even simplified rules cause overfitting because they match surface patterns

2. **Context Size Correlates with Overfitting**: 88KB context still correlates with 14% train-val gap

3. **Negative Learning Has Limits**: Teaching what NOT to do doesn't teach what TO do

**The Solution: Structural Decomposition**

1. **Fact → Act Mapping**: Teach model to extract discrete criminal acts from narrative facts

2. **Act → Charge Matching**: Apply legal definitions to each act separately

3. **Multi-Charge by Structure**: Detect multiple charges through structural markers (numbered facts, temporal markers), not patterns

4. **Minimal Context**: Keep only structural guidance and charge definitions (target $< 30$KB)

---

A key innovation is the introduction of *processing stages* for error diagnosis. The skill categorizes by *where* in the processing pipeline the error occurred, rather than categorizing errors by charge type:

---

**Error Categorization by Processing Stage**

**Processing Stage Failures**:

1. `fact_comprehension_error`: Failed to understand what happened

2. `act_extraction_error`: Failed to identify discrete criminal acts

3. `act_independence_error`: Failed to recognize acts as independent vs. dependent

4. `charge_selection_error`: Applied wrong charge to correctly identified act

5. `charge_name_error`: Wrong or incomplete official charge name

6. `multi_charge_detection_error`: Failed to detect multiple independent charges

---

This represents a meta-learning insight: the skill learns that *how* errors occur matters more than *which* charges are confused.

E.4.3. CONTEXT EVOLUTION

The context evolves into a structurally-organized system with 11 specialized files totaling approximately 30KB (reduced from 88KB in earlier iterations). Unlike pattern-based approaches, the context teaches a systematic *processing framework*:

---

**Case Decomposition Framework (From Context)**

**Process: READ → DECOMPOSE → MATCH → VALIDATE**
**Step 1: Full Case Reading**
Before ANY analysis, read the ENTIRE case description to understand: WHO, WHAT, WHEN, WHERE, HOW, WHY
**Step 2: Act Extraction**
Extract ALL discrete criminal acts. An "act" is a separately described behavior with distinct victim, method, or intent.
**Act Extraction Rules**:

- Numbered subsections ((一)、(二)...) → Separate acts

- Temporal markers (different dates, "随后", "另") → Separate acts

- Different victims/objects → Separate acts

- "另案处理", "另查明" → Additional acts mentioned

**Step 3: Act-to-Charge Matching**
For EACH extracted act: (1) Identify criminal nature, (2) List possible charges, (3) Verify elements satisfied, (4) Select most specific charge

---

The context also includes *structural anti-patterns*, explicit documentation of processing errors to avoid:

> **Structural Anti-Patterns (From Context)**
>
> **Anti-Pattern 1: Merging Numbered Facts**
> **Error**: Treating （一）、（二）as one act
> **Prevention**: Numbered subsections ALWAYS = separate charges
> **Anti-Pattern 2: Ignoring "另案处理"**
> **Error**: Missing additional charges after "另案处理" mentions
> **Prevention**: Scan for THIS defendant's other acts
> **Anti-Pattern 3: Wrong Charge Family**
> **Error**: Predicting wrong charge type by keyword matching
> **Prevention**: Verify ALL elements match before selecting charge

The retrieval function (177 lines) is notably simpler than symptom diagnosis (1,440 lines), reflecting the task's different requirements. Rather than complex routing logic, it implements a *trigger-based augmentation* strategy:

> **Retrieval Function Structure (Overview)**
>
> **Architecture**: 177 lines of Python with trigger-based context augmentation
> **Main Logic**:
>
> 1. **Format Rules**: Always include output format requirements
> 2. **Multi-Charge Triggers**: Detect structural markers ((一), 另案处理, 另查明)
> 3. **High-Error Patterns**: Add guidance for frequently confused charges
> 4. **Charge Name Warnings**: Ensure complete official names are used
> 5. **Core Framework**: Always include the decomposition process
> 6. **Reference Materials**: Append relevant context files

> **Retrieval Function Excerpt**
>
> ```python
> # Multi-charge detection triggers
> multi_charge_triggers = [
>         ("（一）", "Numbered subsection - separate charge"),
>         ("另案处理", "Separate case - check other acts"),
>         ("另查明", "Additional discovery - more charges"),
> ]
> for trigger, guidance in multi_charge_triggers:
>     if trigger in question:
>         relevant_context.append(guidance)
> ```

This design reflects the task's nature: legal classification requires systematic *structural analysis* of case facts rather than semantic pattern matching. The retrieval function ensures the model receives consistent structural guidance plus case-specific warnings, enabling generalization through process, in contrast to pattern accumulation observed in ACE.

## E.5. Aegis2.0

### E.5.1. TASK CHARACTERISTICS AND BASELINE LIMITATIONS

Aegis2.0 requires classifying user prompts into five categories: Copyright Trademark Plagiarism, Political Misinformation Conspiracy, Unauthorized Advice, Immoral Unethical, and safe. We deploy Qwen3-8B as the generator for this task since safety guardrails require small models for efficient inference. This requirement poses unique challenges due to limited reasoning capacity and context window constraints of small models. Therefore, we observe that CE methods with brevity bias, such as GEPA, can outperform the others.

### E.5.2. SKILL EVOLUTION

The initial skill adopts a meta-learning error synthesis approach, categorizing errors by root cause and synthesizing generalized patterns:

**Initial Skill (Excerpt)**

**Error Taxonomy**:

- `false_positive`: Predicted unsafe, should be safe
- `false_negative`: Predicted safe, should be unsafe
- `wrong_category`: Unsafe but wrong category assigned
- `format_error`: Output format or structure issues

**Key Principles**:

1. **Generalize, Don't Overfit**: Synthesize pattern families, not specific examples
2. **Prioritize High-Precision Rules**: Each rule should have clear trigger AND exclusion conditions
3. **Minimal Context, Maximum Clarity**: Prefer rules over examples (small models learn rules better)

After iterations, the skill evolves to emphasize *balanced precision control*. The meta-agent discovers that small models are particularly prone to over-classification (flagging benign prompts as unsafe), and shifts focus to surgical fixes with explicit precision/recall trade-offs:

**Optimal Skill (Excerpt)**

**Core Problem**: Small models (8B parameters) are prone to two critical failures:

1. **Over-classification (False Positives)**: Flagging benign prompts as unsafe

    - Common with: creative roleplay, content templates, conversational statements
    - Root cause: keyword patterns too broad, missing safe exclusions

2. **Under-classification (False Negatives)**: Missing actual violations

    - Common with: content extraction jailbreaks, indirect cheating patterns
    - Root cause: missing trigger patterns, context not retrieved

**Fix Prioritization**:

- If over-classification dominant → Add safe exclusion patterns
- If under-classification dominant → Add missing trigger patterns
- Maintain precision/recall balance score > 0.7

A key innovation is the introduction of *critical distinctions* that small models struggle with:

**Critical Distinctions from Optimal Skill**

**CREATE vs EXTRACT**:

- CREATE new content = SAFE
- EXTRACT/REWRITE existing = Check for plagiarism intent

**Roleplay Detection**:

- Benign roleplay without override = SAFE
- Roleplay with override + harmful content = UNSAFE
- "Stay in character" AND NOT ["no rules", "no ethics"] = SAFE

**Setup-Only Jailbreaks**:

- Jailbreak persona + confirmation request only = SAFE (no harmful action requested)
- Jailbreak persona + action request = UNSAFE

### E.5.3. CONTEXT EVOLUTION

The context evolves into 12 specialized files organized around violation categories and safe exclusions. The context emphasizes *decision rules* over examples, optimized for small model reasoning capacity.

---

**Context File Organization**

**Safe Baseline** (always included):

- `00_safe_baseline.md`: Default safe classification principles
- `10_safe_content_creation.md`: Benign content patterns

**Category-Specific Rules**:

- `01_copyright_patterns.md`: Academic cheating, plagiarism
- `02_misinformation_patterns.md`: Conspiracy, political loaded content
- `03_unauthorized_advice.md`: Medical/legal/financial advice
- `04_immoral_unethical_patterns.md`: Jailbreak personas, override language

**Precision Control** (false positive prevention):

- `06_over_classification_patterns.md`: Patterns to NOT flag
- `07_setup_only_exclusions.md`: Confirmation-only jailbreaks
- `09_professional_roleplay_detection.md`: Benign vs unsafe roleplay

---

The retrieval function (995 lines) implements a *precision-first cascade* with early safe returns for benign patterns:

---

**Retrieval Function Structure (Overview)**

**Architecture**: 995 lines of Python with precision-controlled pattern matching
**Main Logic** (in priority order):

1. **Early Safe Returns**: Detect benign content creation, SEO work, short informal questions → return safe context immediately
2. **Template Jailbreak Detection**: Placeholders + override/anti-detection keywords
3. **Professional Roleplay**: Licensed professionals (doctor/lawyer) vs technical roles
4. **Category-Specific Patterns**: Misinformation, unauthorized advice, immoral content
5. **Over-Classification Prevention**: Add safe exclusion context for ambiguous cases

---

The retrieval function's early safe return mechanism is critical for preventing over-classification:

---

**Early Safe Return Logic (Excerpt)**

```
# Benign content creation patterns
seo_content_patterns = [
    'seo-optimized', 'blog post', 'article',
    'keyword research', 'content strategy'
]
# Short informal questions (< 10 words)
# without specific harm keywords = SAFE
is_short_informal = word_count < 10
    and not has_specific_harm
# Early return for benign patterns
if is_benign_content or is_short_informal:
    return safe_content_context
```

---

This design reflects the unique challenges of safety classification with small models: the retrieval function acts as a *precision gate*, preventing the model from seeing violation-related context when the prompt is clearly benign. This architectural choice reduces over-classification by ensuring the model only receives category-specific rules when patterns genuinely

warrant scrutiny.

## F. Utility Functions

The utility functions to call LLMs (fixed to DeepSeek V3.1) and embedding models (fixed to OpenAI's text-embedding-3-small) are provided below.

**utils/llm.py**

```python
from langchain_openai import ChatOpenAI
from typing import List, Type, TypeVar, Union
from pydantic import BaseModel, Field
import asyncio
import os

from dotenv import load_dotenv

load_dotenv(override=True)

T = TypeVar('T', bound=BaseModel)

MAX_CONCURRENCY = 50
MAX_LLM_CALLS = 100

llm = ChatOpenAI(
    model=os.getenv("SANDBOX_MODEL"),
    api_key=os.getenv("OPENROUTER_API_KEY"),
    base_url=os.getenv("OPENROUTER_API_BASE"),
    temperature=0,
)

class TextResponse(BaseModel):
    """Simple text response from LLM."""
    response: str = Field(description="The LLM's response text")

async def call_llm_async(
    prompts: List[str],
    schema: Type[BaseModel],
) -> List[T]:
    """
    Call llms with structured output

    Args:
        prompts: List of prompts to send to the LLM
        schema: Pydantic BaseModel class defining the output structure

    Returns:
        List of instances of the schema class with LLM outputs

    Raises:
        ValueError: If batch limit is exceeded
    """
    if len(prompts) > MAX_LLM_CALLS:
        raise ValueError(f"Number of prompts ({len(prompts)}) exceeds maximum allowed
    per batch ({MAX_LLM_CALLS})")

    llm_with_structure = llm.with_structured_output(schema).with_retry(
    stop_after_attempt=3)
    semaphore = asyncio.Semaphore(MAX_CONCURRENCY)

    async def process_single(prompt: str) -> T:
```

```python
        """Process a single prompt with semaphore control."""
        async with semaphore:
            result = await llm_with_structure.ainvoke(prompt)
            return result

    results = await asyncio.gather(*[process_single(prompt) for prompt in prompts])
    return results

def call_llm(
    prompts: Union[str, List[str]],
    schema: Type[BaseModel] = None,
) -> Union[str, List[str], BaseModel, List[BaseModel]]:
    """
    Synchronous wrapper for LLM calls. Supports both single and batch prompts.

    Args:
        prompts: Single prompt string or list of prompts
        schema: Optional Pydantic BaseModel class. If None, returns plain text
    responses.

    Returns:
        - If prompts is a string and schema is None: returns string
        - If prompts is a string and schema is provided: returns schema instance
        - If prompts is a list and schema is None: returns list of strings
        - If prompts is a list and schema is provided: returns list of schema instances

    Examples:
        # Simple text response
        response = call_llm("What is 2+2?")
        print(response)  # "4"

        # Batch text responses
        responses = call_llm(["What is 2+2?", "What is 3+3?"])
        print(responses)  # ["4", "6"]

        # Structured response
        class Analysis(BaseModel):
            pattern: str
            confidence: float

        result = call_llm("Analyze this...", schema=Analysis)
        print(result.pattern)

        # Batch structured responses
        results = call_llm(["Analyze A", "Analyze B"], schema=Analysis)
        for r in results:
            print(r.pattern)
    """
    is_single = isinstance(prompts, str)
    prompt_list = [prompts] if is_single else prompts
    use_schema = schema if schema is not None else TextResponse
    results = asyncio.run(call_llm_async(prompt_list, use_schema))
    if schema is None:
        results = [r.response for r in results]
    return results[0] if is_single else results
```

**utils/embedding.py**

```python
from typing import List
import numpy as np
```

```python
from numpy.typing import NDArray
from langchain_openai import OpenAIEmbeddings
import os

from dotenv import load_dotenv

load_dotenv(override=True)

EMBEDDING_MODEL = "text-embedding-3-small"

embeddings = OpenAIEmbeddings(
    model=EMBEDDING_MODEL,
    api_key=os.getenv("OPENROUTER_API_KEY"),
    base_url=os.getenv("OPENROUTER_API_BASE"),
)

def compute_embedding_similarity(
    strings_a: List[str],
    strings_b: List[str],
) -> NDArray[np.float64]:
    """
    Compute cosine similarity between embeddings of two lists of strings.

    Args:
        strings_a: First list of strings
        strings_b: Second list of strings

    Returns:
        A 2D numpy array of shape (len(strings_a), len(strings_b)) containing
        cosine similarity scores between each pair of strings.
    """

    # Get embeddings for both lists
    embeddings_a = embeddings.embed_documents(strings_a)
    embeddings_b = embeddings.embed_documents(strings_b)

    # Convert to numpy arrays
    embeddings_a_np = np.array(embeddings_a)
    embeddings_b_np = np.array(embeddings_b)

    # Compute cosine similarity
    # Normalize the embeddings
    embeddings_a_norm = embeddings_a_np / np.linalg.norm(embeddings_a_np, axis=1,
    keepdims=True)
    embeddings_b_norm = embeddings_b_np / np.linalg.norm(embeddings_b_np, axis=1,
    keepdims=True)

    # Compute dot product (cosine similarity for normalized vectors)
    similarity_matrix = embeddings_a_norm @ embeddings_b_norm.T

    return similarity_matrix
```

