# OpenReview forum: "Meta Context Engineering via Agentic Skill Evolution"
_ICML.cc/2026/Conference — ICML 2026 regular_

### Official Review · Reviewer_AuDs · 2026-03-08

**Soundness:** 3
**Presentation:** 3
**Significance:** 3
**Originality:** 3
**Overall Recommendation:** 5
**Confidence:** 4

**Summary:**

Existing Context Engineering (CE) approaches are constrained by manually designed fixed structures, and limit the design space for context optimization. This paper proposes Meta Context Engineering (MCE), a bi-level optimization framework that co-evolves CE skills and context artifacts through "agentic crossover," representing context as flexible code and files rather than fixed schemas.

**Compliance With Llm Reviewing Policy:**

Affirmed.

**Key Questions For Authors:**

How significant is the impact of skill initialization on the algorithm's stability and convergence behavior?

**Limitations:**

yes

**Strengths And Weaknesses:**

**Strengths:**

1. Treating "skills" as a novel abstraction layer for evolutionary computation, different from traditional prompt or workflow optimization. This is a shift from fixed workflows toward autonomous agent systems.

2. Formalizing context engineering as a bi-level optimization problem that separates meta-level skill optimization from base-level context optimization; the algorithm appears sound.

3. Experimental validation across diverse domains encompassing both offline and online settings, demonstrating the method's generality.

**Weaknesses:**

I don't identify significant weaknesses.

---

> ### Author Rebuttal · Authors · 2026-03-30
>
> Thank you for the careful evaluation of our work, and for recognizing the novelty of skills as an abstraction layer for evolutionary CE, the soundness of the bi-level formulation, and the breadth of the empirical validation.
>
> > How significant is the impact of skill initialization on the algorithm’s stability and convergence behavior?
>
> We conducted supplementary multi-seed runs on Symptom2Disease under a reduced budget relative to the main experiments, namely 200 total rollouts (`K=4`), for two backbone variants, `minimax-m2.1` and `minimax-m2.7`. With three independent workspace seeds per model and the default skill initialization, we report the best validation accuracy. This yields 85.7 ± 0.3% for `minimax-m2.1` and 85.7 ± 1.9% for `minimax-m2.7` (mean ± sample standard deviation, n=3).
> Overall, the cross-seed spread remains moderate for both backbones, suggesting that default initialization is not the dominant factor here: runs stay within a similar accuracy band when skills are bootstrapped by the same backbone model and meta-agent setup.

---

> > ### Author Rebuttal · Reviewer_AuDs · 2026-04-03
> >
> > The author's response is clear for answering my question.

---

> > > ### Author Response · Authors · 2026-04-06
> > >
> > > Thanks again for your review.

---

### Official Review · Reviewer_2JUH · 2026-03-13

**Soundness:** 2
**Presentation:** 3
**Significance:** 3
**Originality:** 3
**Overall Recommendation:** 4
**Confidence:** 3

**Summary:**

This paper introduces Meta Context Engineering (MCE), a bi-level optimization framework for LLM inference-time context. The meta-level evolves CE skills (folders of instructions, code, and templates) via agentic crossover—an LLM-driven operator that reasons over skill history and performance metrics. The base-level executes these skills to optimize context as unconstrained files and code from training rollouts. Across five classification/tagging benchmarks (finance, chemistry, medicine, law, AI safety), MCE achieves 89.1% average relative improvement over the base model in the offline setting (vs. 70.7% for ACE), while demonstrating adaptive context length (1.5K–86K tokens), 13.6× training speedup, and lower degradation under strong-to-weak context transfer.

**Compliance With Llm Reviewing Policy:**

Affirmed.

**Final Justification:**

After thorough consideration, I have decided to maintain my score.

**Key Questions For Authors:**

- Can you run MCE with DeepSeek V3.1 as the sole model for both meta and base agents? If MCE still outperforms ACE under identical model access, this would decisively address W1.
- What happens if the three example skill templates are removed from the meta-agent prompt? This would disentangle the bi-level framework's contribution from the pre-encoded CE knowledge.
- How sensitive is performance to K? Can you provide learning curves (K=1,...,10) and compare against random skill sampling without history access on at least two benchmarks? This would validate whether agentic crossover provides value beyond multiple attempts.

**Limitations:**

yes

**Strengths And Weaknesses:**

**Strengths**
- The analysis of brevity bias (GEPA) vs. verbosity bias (ACE) is convincing and empirically grounded—GEPA excels on LawBench where conciseness helps but fails on USPTO requiring detailed rules; ACE outperforms on FiNER but underperforms even ICL on Symptom2Disease. Decoupling "how to engineer context" from "what context to produce" via bi-level optimization is a natural and insightful formalization.
- The paper systematically analyzes context length adaptability, token efficiency, strong-to-weak transferability, and training efficiency (13.6× speedup, 4.8× fewer rollouts). These complementary analyses collectively build a compelling case.
- The appendix case studies reveal genuinely interesting emergent behaviors: on FiNER, skills evolve from pattern extraction to error-driven generalization with anti-patterns; on Symptom2Disease, the meta-agent learns when to stop optimizing; on LawBench, skills shift from pattern matching to structural case decomposition. These demonstrate the meta-level discovers strategies beyond hand-designed heuristics.


**Weaknesses**
- MCE uses MiniMax M2.1 as the agentic model plus DeepSeek V3.1 as a callable utility within the sandbox. Baselines use only DeepSeek V3.1. The Appendix B ablation shows MiniMax degrades ACE's reflector/curator role, but this does not rule out its contribution in MCE's fundamentally different paradigm (writing code, manipulating files, designing retrieval functions).
- The system prompt provides three example skill templates (including an explicit "ACE-Style Reflection & Curation" template), effectively pre-encoding multiple CE strategies. This partially undermines the claim of "replacing heuristic scaffolding with a generic design space"—discovery of effective strategies is partly recombination of pre-encoded knowledge.
- The authors acknowledge limitations on reasoning-intensive tasks, but this is understated—the entire empirical validation falls within MCE's acknowledged sweet spot. No generation, multi-step reasoning, planning, or open-ended tasks are evaluated, significantly limiting generalizability claims.
- K=5 means only 5 skills are evaluated. No analysis of sensitivity to K is provided, nor is there a comparison against random skill sampling without history access. It remains unclear whether improvements stem from intelligent crossover or simply from having multiple attempts.

---

> ### Author Rebuttal · Authors · 2026-03-30
>
> Thank you for your careful evaluation of our work, and for recognizing the paper’s core strengths, including the bi-level formalization, the brevity/verbosity analysis, the efficiency and transferability results, and the appendix case studies.
>
> > W1: MCE uses MiniMax M2.1 as the agentic model plus DeepSeek V3.1 as a callable utility, while baselines use only DeepSeek V3.1.
>
> MCE is fundamentally enabled by—and scales with—the agentic capabilities of advanced foundation models. Because we observed that DeepSeek V3.1 struggled to produce meaningful optimizations within our agent harness (Claude Agent SDK), we here additionally evaluated both MCE and ACE using MiniMax M2.1 and M2.7 as the sole models. For MCE, we report the results using two prompt variants: one with three example skill templates, and one without (see W2).
>
> Due to rebuttal time constraints, we use a reduced budget relative to the main experiments: symptom diagnosis task, 50 train/val rollouts per iteration, and K=4.
>
> | Sole model | MCE | ACE | Base Model |
> | --- | --- | --- | --- |
> | MiniMax M2.1 | 85.8% / 85.8% | 75.4% | 66.5% |
> | MiniMax M2.7 | 85.4% / 87.7% | 73.1% | 70.3% |
>
> The core idea behind MCE—and why we believe it is essential to evaluate it using strong agentic models—is that agent harness research must adapt to and build upon the rapid progress of underlying foundation models. While ACE may be better suited for weaker models that lack sufficient agency, MCE demonstrates how the growing capabilities of agentic foundation models enable novel paradigms of autonomous self-evolution.
>
> > W2: The system prompt provides three example skill templates, partially undermining the claim of replacing heuristic scaffolding with a generic design space.
>
> Under the same experimental setting as W1, we ablate the three example skill templates in the meta-agent prompt.
>
> | Model | Meta prompt w/ 3 example templates | Meta prompt w/o example templates |
> | --- | --- | --- |
> | MiniMax M2.1 | 85.8% | 85.8% |
> | MiniMax M2.7 | 85.4% | 87.7% |
>
> The results show that the examples are optional steering that does not necessarily help.
>
> > W3: The empirical validation falls within MCE’s acknowledged sweet spot; no generation, multi-step reasoning, planning, or open-ended tasks are evaluated.
>
> We have additionally conducted experiments on HotpotQA (train/val/test: 150/50/50), which features multi-hop reasoning, planning of agentic retrieval, and open-ended question answering (which causes many formatting issues).
> We compare MCE (MiniMax M2.1, agentic setting, K=3, 250 total train/val rollouts) against GEPA (MiniMax M2.1, its original open-source setting, 1000 total rollouts). The results (F1) are shown below:
>
> | Method | Base Model | Optimized |
> | --- | --- | --- |
> | GEPA | 0.47 (workflow) | 0.56 |
> | MCE | 0.63 (agent) | 0.78 |
>
> The results show that, in a fully agentic setup (context as files in the agent workspace; context interface is not needed), MCE achieves better base and optimized performance than GEPA in its open-source workflow setting. It supports the extensibility of MCE.
>
> > W4: No sensitivity to K is provided, nor a comparison against random skill sampling without history access.
>
> Using the same experimental setup as in W1, we evaluate MCE's sensitivity to the number of outer iterations K and compare it against a random skill sampling baseline.
>
> The table below reports the cumulative maximum validation accuracy up to iteration K. For comparison, we include 1) a baseline that randomly samples skills (with 1-epoch optimization per skill) and 2) a 1-epoch ACE run with a compute budget comparable to K=5.
>
> | K | MCE with Skill Sampling | MCE | ACE |
> | --- | --- | --- | --- |
> | 1 | 77.36% | 78.77% | - |
> | 3 | 82.08% | 83.02% | - |
> | 5 | 82.08% | 85.85% | 75.4% (1 epoch ACE) |
> | 10 | 83.49% | 86.79% | - |
>
> Results indicate that while random skill sampling quickly plateaus, MCE's history-informed skill evolution drives continuous improvement. By explicitly learning from past failures and evolution trajectory, the evolutionary process helps the model correct historical mistakes and mitigate overfitting to the training data (see Appendix E for case studies).
>
>
> > Q1: Can you run MCE with DeepSeek V3.1 as the sole model for both meta and base agents?
>
> Please refer to our response to W1.
>
> > Q2: What happens if the three example skill templates are removed from the meta-agent prompt?
>
> Please refer to our response to W2.
>
> > Q3: How sensitive is performance to K?
>
> Please refer to our response to W4.

---

> > ### Author Rebuttal · Reviewer_2JUH · 2026-04-03
> >
> > Thank you for the rebuttal. I think most of my concerns have been addressed. I will keep my positive score.

---

> > > ### Author Response · Authors · 2026-04-06
> > >
> > > Thank you again for your review.

---

### Official Review · Reviewer_QUmo · 2026-03-13

**Soundness:** 3
**Presentation:** 3
**Significance:** 3
**Originality:** 3
**Overall Recommendation:** 5
**Confidence:** 4

**Summary:**

The paper studies context engineering (CE) for LLMs, which optimizes inference-time context rather than model parameters. It proposes Meta Context Engineering (MCE), a bi-level framework where a meta-agent evolves context-engineering skills while a base-agent executes these skills to construct context artifacts (represented as files and code). The meta-agent performs evolutionary search over skills using an LLM-based crossover operator, while the base-agent optimizes the context using training rollouts. Experiments across five domain benchmarks compare MCE with existing CE approaches such as ACE, GEPA, and MIPROv2. Results show consistent improvements in task performance, context efficiency, and training efficiency.

**Compliance With Llm Reviewing Policy:**

Affirmed.

**Final Justification:**

My questions and concerns have been satisfactorily addressed by the authors during the rebuttal process. I appreciate the additional clarifications and supporting evidence provided, which have strengthened my understanding of the work. I have increased my score to 5.

**Key Questions For Authors:**

1. The paper notes that MCE may not benefit reasoning-intensive tasks. Have the authors evaluated the approach on reasoning-heavy benchmarks (e.g., math or multi-step planning tasks)?
2. Since skills can include arbitrary instructions, code, and files, it is unclear how large or structured the optimization space is. Clarifying how the search is constrained or guided would help assess whether the framework is practically tractable. If the authors can show that the search space is effectively regularized, it would strengthen the methodological contribution.
3. How does the proposed agentic crossover differ from existing LLM-based evolutionary operators? The paper describes crossover as an LLM agent reasoning over past skills and recombining them, but the distinction from prior LLM-driven evolutionary search methods is not fully clear. A clearer explanation or comparison would help assess the novelty of the method.
4. What kinds of skills are actually discovered during evolution? The paper would benefit from concrete examples and analysis of evolved skills and how they differ from existing CE pipelines (e.g., ACE-style workflows). If the discovered skills demonstrate qualitatively new strategies, this would strengthen the claim that the framework explores a broader design space.
5. How sensitive is the framework to the choice of agent model used for skill evolution? Since the meta-agent plays a central role in generating new skills, it would be helpful to understand whether the performance depends heavily on the capability of the specific agent model used. If the method remains effective across different agent models, it would improve confidence in its general applicability.

**Limitations:**

Yes.

**Strengths And Weaknesses:**

### Strengths
1. The paper formalizes context engineering as a bi-level optimization problem, which provides a clean conceptual framework.
2. Experiments cover five different benchmarks, suggesting the approach is not limited to a single task.
3. The proposed method consistently outperforms several recent CE baselines in the reported experiments.
4. Interesting direction. Treating context engineering strategies as evolvable agent skills is a novel perspective that may inspire further work on automated LLM pipeline design.

### Weaknesses
1. The framework includes several components (meta-agent, base-agent, skill evolution, agent tools), making it difficult to determine which part drives the improvements. Attribution of gains is unclear.
2. Limited ablation analysis. The ablation study is relatively narrow and mostly conducted on a single dataset.
3. The proposed optimization ultimately relies on a simple evolutionary strategy, and the crossover operation is implemented through an LLM agent. Conceptually, this remains close to existing LLM-driven evolutionary search frameworks.
4. The notion of a “skill” (a folder containing instructions, scripts, and resources) is flexible but not formally specified, making the optimization space difficult to characterize and analyze.

---

> ### Author Rebuttal · Authors · 2026-03-30
>
> Thank you for your careful evaluation and for the recognition.
>
> > W1 and W2: Inadequate ablation
>
> Agentic foundation models (with tools and computer access) provide the capability envelope, while our method contributes task framing and scaffolding for the bi-level loop. Consistent with our current ablations, the observed improvements are attributable to the combination of meta-level CE-skill evolution and base-level agentic context optimization.
>
> In Table 3, MCE (w/o skills) vs. ACE verifies the improvements from base-level optimization; MCE variants validate the contributions from meta-level skill evolution. We extend the ablation to an additional task (Symptom2Disease) and report the results below, and the pattern is consistent with Table 3.
>
> | Method | Offline | Online |
> | --- | ---: | ---: |
> | Base model | 63.7 | - |
> | ACE | 79.2 | 62.3 |
> | MCE (w/o skills) | 84.4 | 76.9 |
> | MCE (w/ a fixed skill) | 83.0 | 76.4 |
> | MCE (full, w/ evolving skills) | 89.2 | - |
>
> > W3: The used ES remains close to existing frameworks.
>
> The high-level algorithm is intentionally simple; the novelty lies in *what* is optimized and *how* the operator is instantiated.
>
> 1. **Target/Encoding.** Prior LLM-driven evolutionary methods typically optimize prompts, programs, or candidate outputs under relatively fixed context representations. MCE instead optimizes CE skills: higher-order procedures that determine how context is represented, learned, and retrieved, with the base agent free to realize them as files, code, and retrieval logic.
> 2. **Method/Operator.** Agentic crossover exposes full history through a filesystem, enabling selective diagnosis of raw prior code and execution traces rather than optimization from compressed per-candidate summaries. This opens a broader search space than fixed recombination over text or program candidates, and suggests a promising direction for training more general agentic capabilities around such operators.
>
> > W4: The notion of a “skill” makes the optimization space difficult to characterize and analyze.
>
> Following Anthropic's definition, we use skill in a practical sense: a directory of instructions, scripts, and resources that helps an agent perform a task.
>
> We acknowledge the dual nature of treating such a broad abstraction as an optimization target. On one hand, a highly general representation makes the optimization space inherently difficult to formally characterize and analyze. On the other hand, we intentionally maintain this broad and encompassing formulation because skills must be versatile. Depending on the task, an effective skill may rely on natural-language methodology, executable code, retrieval functions, or reference materials. Imposing a narrow formal schema would exclude useful forms of agentic capability and reintroduce the representation bias that MCE is designed to overcome.
>
> To mitigate the challenges of navigating this complex space, the search in MCE is regularized by the task specification, restricted workspace access, the requirement to produce a callable retrieval interface, warm-starting from the current best context, and validation-driven selection.
>
>
> > Q1: Evaluation on reasoning-heavy benchmarks
>
> Our existing suite requires domain-specific reasoning: e.g., Symptom2Disease involves clinical differential diagnosis; FINER demands fine-grained semantic disambiguation; and USPTO-50k requires multi-step chemical retrosynthesis reasoning (analyzing functional groups, reaction mechanisms, and bond-breaking strategies).
>
> In addition, we have conducted new experiments on HotpotQA (train/val/test: 150/50/50), a benchmark designed to test multi-hop reasoning. We compare MCE (MiniMax M2.1, agentic setting, K=3, 250 total rollouts) against GEPA (MiniMax M2.1, its open-source workflow setting, 1000 total rollouts). The results (F1) are shown below:
>
> | Method | Base Model | Optimized |
> | --- | --- | --- |
> | GEPA | 0.47 (workflow) | 0.56 |
> | MCE | 0.63 (agent) | 0.78 |
>
> The results show that MCE, with a fully agentic setup, can achieve better base and optimized performance than GEPA.
>
> > Q2: How is the search constrained?
>
> Please refer to our response to W4.
>
> > Q3: How does agentic crossover differ from prior operators?
>
> Please refer to our response to W3.
>
> > Q4: What kinds of skills are discovered during evolution?
>
> We present and discuss the skills evolved in Appendix D.
>
> > Q5: How sensitive is the framework to the agent model used for skill evolution?
>
> We evaluate three models (MiniMax M2.1, MiniMax M2.7, and Claude Opus 4.6) on Symptom2Disease under a reduced budget (K = 4, 200 training rollouts). Performance scales with model capability, showing that MCE harness can continually leverage model improvements.
>
> | Model | MCE |
> | --- | --- |
> | MiniMax M2.1 | 85.8% |
> | MiniMax M2.7 | 87.7% |
> | Claude Opus 4.6 | 88.0% |

---

> > ### Author Rebuttal · Reviewer_QUmo · 2026-04-02
> >
> > The rebuttal addresses several of my concerns and adds useful new evidence. The extended ablations (e.g., Symptom2Disease) and the additional HotpotQA results make the empirical story stronger. I also appreciate the clarification on the roles of base-level optimization vs. meta-level skill evolution.
> >
> > I still have some reservations on the conceptual side:
> >
> > - For W3, I see the intended distinction (optimizing skills vs. prompts/programs), but the difference from prior LLM-based evolutionary or agent optimization approaches still feels somewhat high-level. It would help to more concretely pin down what makes the search space and operator fundamentally different.
> > - For W4, I understand the motivation for keeping the notion of “skills” broad, but this also makes the optimization space hard to characterize. I’m still a bit unclear on how this affects reproducibility and stability in practice.
> >
> > Overall, the rebuttal improves my confidence in the empirical results, but I think some of the core conceptual questions remain open.

---

> > > ### Author Response · Authors · 2026-04-07
> > >
> > > Thank you for your continued engagement. We respond to the two remaining concerns below.
> > >
> > > > W3 (follow-up): The difference from prior LLM-based evolutionary or agent optimization approaches still feels somewhat high-level. It would help to more concretely pin down what makes the search space and operator fundamentally different.
> > >
> > > The key difference is to expose full iteration history through the workspace filesystem, enabling selective diagnosis of prior skills and context artifacts rather than optimization from compressed per-candidate summaries.
> > >
> > > The meta-agent is a coding agent that can invoke tools and read/write files. This choice of coding agent (rather than a raw LLM call) matters because the accumulated experience across iterations quickly exceeds context limits; the meta-agent must decide what to inspect and validate edits through direct interaction with the workspace. In practice, at each crossover step the meta-agent reads the skills from prior iterations, inspects the `context/` artifacts (taxonomy, confusion pairs, concrete examples) that the base agent built up, and parses per-example train/validation outcomes—then synthesizes a new skill that addresses observed failure modes. This is precisely what prior text optimizers cannot do: OPRO and AlphaEvolve condition only on scalar scores; TextGrad on a single textual gradient; GEPA on a per-candidate reflective summary. MCE's workspace stores the full diagnostic footprint of every iteration—skills, learned artifacts, and per-example evaluation results—which the meta-agent retrieves via standard file operations rather than ingesting as a single prompt.
> > >
> > > > W4 (follow-up): The broad definition of "skill" makes the optimization space hard to characterize; it is still unclear how this affects reproducibility and stability in practice.
> > >
> > > We find that three mechanisms keep the effective search space well-behaved in practice.
> > >
> > > First, **structural regularization**: every skill follows a partially fixed schema (YAML frontmatter fields and a Skill Overview section in `SKILL.md`), which constrains the representation without fully specifying its content.
> > > Second, **validation-driven evaluation**: every proposed skill is assessed against a held-out validation set before the meta-agent proceeds, making skill quality directly observable and comparable across iterations; the meta-agent reads all prior validation outcomes from the workspace and uses them to identify which directions worked and which failed.
> > > Third, **agent-side inductive bias**: a capable coding agent naturally proposes internally coherent procedures rather than arbitrary file collections.
> > >
> > > Empirically, three independent seeds (see our response to Reviewer AuDs, Q1) on Symptad2Disease yield 85.7 ± 0.3% (MiniMax M2.1) and 85.7 ± 1.9% (MiniMax M2.7)—low variance despite the open-ended representation. Furthermore, MCE scales reasonably with backbone capability (Q5).
> > >
> > > ----
> > >
> > > Thanks again for your dedicated review.

---

### Official Review · Reviewer_vnhb · 2026-03-13

**Soundness:** 3
**Presentation:** 3
**Significance:** 3
**Originality:** 3
**Overall Recommendation:** 4
**Confidence:** 3

**Summary:**

The paper introduces a bi-level optimization framework (MCE) for learning how to construct and optimize context for LLMs during inference. It co-evolves two components: context engineering skills which encode how the context should be represented and optimized, and context artifacts. At the meta-level, MCE checks the history of previous skills and the resulting context functions with validation metrics to synthesize an improved skill. At the base-level, an agent executes the current skill to produce a context function by analyzing the rollouts and updating context files.

**Compliance With Llm Reviewing Policy:**

Affirmed.

**Final Justification:**

I thank the authors for their rebuttal and raised my confidence score.

**Key Questions For Authors:**

See the weaknesses in Strengths And Weaknesses.

**Limitations:**

Yes.

**Strengths And Weaknesses:**

Strengths:
- The problem is well-motivated with clear formulation and overall algorithm.
-	The meta-optimization with agentic crossover is well-designed compared with applying fixed recombination rules, and the meta-agent inspects and recombines components from previous skill folders allowing it to discover new context engineering workflows.
-	The training efficiency reported in the paper is striking, stemming from the batch processing in the base-agent.

Weaknesses:
- The agentic crossover is conceptually described but under-specified, making it unclear to reason about its behavior.
-	Although the paper explicitly acknowledges MCE may not offer advantages on reasoning-intensive tasks, it is a significant scope restriction, and evaluating on reasoning benchmarks or code generation tasks could better clarify the limits of MCE and allow fairer comparison with the broader context engineering scope.
-	The total API cost of MCE is not reported given that it uses heavy LLM calls per iteration.
-	It is unclear why evolving skills underperforms the skill-less variants in the online setting in Table 1.

---

> ### Author Rebuttal · Authors · 2026-03-30
>
> Thank you for your careful evaluation of our work, and for recognizing the problem formulation, the overall algorithm, the agentic crossover design, and the strong training-efficiency results.
>
> > W1: The agentic crossover is conceptually described but under-specified.
>
> Agentic crossover is a history-informed, validation-driven skill synthesis operator. At iteration \(k\), the meta-agent conditions on three concrete signals: task specification, iteration history (prior skills and learned context artifacts), and train/validation outcomes. The output is a new self-contained skill that explicitly addresses observed failure modes such as overfitting, underfitting, poor structure, or inefficient retrieval.
>
> To give a concrete example from one real crossover trajectory:
>
> [STEP 1] Ingest specification. Read the meta prompt: evolve a transferable `SKILL.md` for the base agent and emit it under the new child workspace.
>
> [STEP 2] Read prior skills. Open the final `SKILL.md` from `iter1_sub3`, `iter2_sub3`, and `iter3_sub3` to trace the progression from taxonomy to error-driven refinement to confusion-pair / anti-pattern guidance.
>
> [STEP 3] Scan learned artifacts. Inspect each iteration's `context/`: early runs contain only a taxonomy file, while later runs add `SUMMARY.md`, `confusion_pairs.md`, and `concrete_examples.md`.
>
> [STEP 4] Read fitness signals. Parse `data/train.json` for aggregate accuracy and per-example correctness to identify persistent failures.
>
> [STEP 5] Spot-check learned rules. Sample the taxonomy, confusion-pair, and example files to see what the base agent actually learned and reused.
>
> [STEP 6] Perform crossover. Keep what helped (taxonomy plus targeted fixes), remove what overfit, and address unresolved confusion pairs by committing to a new category-first skill with stronger disambiguation checks.
>
> [STEP 7] Emit the child. Create the next skill directory and write the new `SKILL.md`.
>
> [STEP 8] Verify structure. Run lightweight checks on files, section headers, and completeness before handing the child skill to the next base-agent run.
>
> > W2: Although the paper acknowledges that MCE may not offer advantages on reasoning-intensive tasks, this is a significant scope restriction.
>
> We observe that existing manually crafted agentic harnesses are highly optimized and well-suited for such tasks. We hypothesized that on such saturated tasks, a general meta-optimization framework like MCE might show less relative gains compared to these heavily hand-engineered pipelines.
>
> However, this does not restrict MCE's scope. Our existing evaluation suite already encompasses tasks requiring domain-specific reasoning (e.g., clinical differential diagnosis in Symptom2Disease, multi-step chemical retrosynthesis in USPTO-50k, and legal element mapping in Crime Prediction).
>
> We conducted new experiments on HotpotQA (train/val/test: 150/50/50), a benchmark designed to test explicit multi-hop reasoning. We compare MCE (MiniMax M2.1, agentic setting, K=3, 250 total train and validation rollouts) against GEPA (MiniMax M2.1, its original workflow setting, 1000 total rollouts). The results (F1) are shown below:
>
> | Method | Base Model | Optimized |
> | --- | --- | --- |
> | GEPA | 0.47 (workflow) | 0.56 |
> | MCE | 0.63 (agent) | 0.78 |
>
> This shows that MCE is capable of handling reasoning-intensive domains, and its skill-search mechanism can effectively navigate these spaces.
>
> > W3: The total API cost of MCE is not reported.
>
> Total API cost depends on the underlying provider and pricing. Our runs go through OpenRouter, which may route requests to different providers, so any figure we report from our logs is necessarily approximate.
>
> In our current setup, both meta-agent and base-agent operations use MiniMax M2.1, and each such operation typically costs on the order of `$`0.1-0.5, depending on the current iteration, task complexity, and context size. The rollouts use DeepSeek V3.1. As a rough reference, on FiNER with roughly 1200 rollouts, the DeepSeek cost is about `$`5-10 in total, depending largely on the learned context length. Overall, an MCE run is therefore approximately `$`5-15 in our logs.
>
>
> > W4: It is unclear why evolving skills underperform the skill-less variants in the online setting in Table 1.
>
> The online setting does not allow the iterative meta-level evolution we use offline (line 730). In the online table, `MCE` employs a fixed skill produced from the task specification only, while `MCE (w/o skills)` is a fully autonomous base agent without that guidance. The ranking therefore reflects how much a fixed skill helps per task, not whether evolution is useful. The observation is consistent with SkillBench [1], where model-written skills (in one shot) often fail to help. That is precisely why meta-level skill evolution (offline) matters.
>
> [1] Li et al. (2026). SkillsBench: Benchmarking how well agent skills work across diverse tasks. arXiv preprint arXiv:2602.12670.

---

> > ### Author Rebuttal · Reviewer_vnhb · 2026-04-02
> >
> > Thanks for your rebuttal. The crossover trace in W1 meaningfully clarifies the operator's behavior and addresses my concern. On W2, the new hotpotQA results are encouraging, but multi-hop QA is not representative of reasoning-intensive tasks such as math or code generation. I will raise my confidence score.

---

> > > ### Author Response · Authors · 2026-04-07
> > >
> > > Thank you for your continued engagement. We verify MCE's benefits on two additional tasks.
> > >
> > > For formal mathematical reasoning, we evaluate on miniF2F-Dafny, where the model must complete Dafny proof bodies verified by a formal checker. MCE improves performance from 26.67% to 46.67% (+20.00 points, 100 rollouts, train/val/test: 100/30/30).
> > >
> > > For code generation, we evaluate on DS-1000, a data science benchmark where correctness is determined by executing test cases. MCE improves performance from 73.33% to 86.67% (+13.34 points, 100 rollouts, train/val/test: 100/30/30).
> > >
> > > Together, these results demonstrate that MCE yields consistent gains across both formal mathematical reasoning and code generation.
> > >
> > > ---
> > >
> > > Thanks again for your dedicated review.

---

### Decision · Program_Chairs · 2026-04-30

**Decision:**

Accept (regular)

**Comment:**

This paper proposes Meta Context Engineering (MCE), a bi-level framework where a meta-agent evolves context-engineering skills and a base-agent uses them to build context functions as files/code for inference-time adaptation. It outperforms prior CE methods across multiple domains and shows gains in performance. The reviewers agree that the paper offers a novel CE abstraction—optimizing skills and context systems rather than fixed workflows, and supports it with strong, broad experiments. The rebuttal addressed the main doubts by showing gains are not just from stronger models, not from the example templates, and not from random multi-try search. The final reviews are 4 / 5 / 4 / 5 and no negative reviewer remaining. In my opinion, I think the paper's impact may be limited by the fact that context function in many harnesses today are implicit decided by the models automatically on the fly, rather than through explicit composition of operators, although I think explicit composition of operators is a good way to clearly make it optimizable. Overall I think this is a clear accept.